# ADAPT: ADAPTIVE PROMPT TUNING FOR PRE-TRAINED VISION-LANGUAGE MODELS

## ABSTRACT

Prompt tuning has emerged as an effective way for parameter-efficient fine-tuning. Conventional deep prompt tuning inserts continuous prompts of a fixed context length into the input to each layer. When a pre-trained model is tailored to a specific downstream task, different layers initialized with pre-trained weights might have, depending on the distribution shift type, different levels of deviation from the optimal weights. Inserted prompts with a fixed context length might have redundant context tokens or insufficient context length. To address this issue, we propose a deep continuous prompting method dubbed Adapt that encourages heterogeneous context lengths. Context lengths are automatically determined by iteratively pruning context tokens. We use the saliency criterion for the neural network pruning to compute the importance scores of context tokens in order to determine which tokens to prune. We examine the proposed method on the pre-trained vision-language model CLIP. Extensive experiments on 11 downstream datasets reveal the advantage of Adapt: the average test accuracy increases from 79.83% to 81.70%. The highest performance gain on individual datasets is 9.63%. At the same time, the computational overheads are comparable to or smaller than baseline methods. We release the code in
`https://anonymous.4open.science/r/Adapt-Release`.

## 1 INTRODUCTION

Large-scale models have gained significant attention in language (Brown, 2020; Wang et al., 2021; Touvron et al., 2023), vision (He et al., 2022; Zou et al., 2024; Esser et al., 2024) and multimodality (Radford et al., 2021; Lu et al., 2022; Lai et al., 2024; Liu et al., 2024). When applying pre-trained large-scale models to various downstream tasks, the zero-shot performance can be sub-optimal. Although fine-tuning remarkably elicits the potential of pre-trained models, the fine-tuning process is computationally expensive. Parameter-efficient fine-tuning (PEFT) offers an efficient way to adapt the pre-trained model to various downstream tasks at a low cost. PEFT enhances the performance of pre-trained models by reparametrizing model weights (Hu et al., 2021; Dettmers et al., 2024; Zhao et al., 2024), additive modules (Chen et al., 2022b; Zhang et al., 2023; Mou et al., 2024) and selective weight updates (Ding et al., 2023; Lawton et al., 2023; Fu et al., 2023). Among PEFT methods, prompting methods have the least effect on backbone models as they focus on the input instead of model parameters.

Prompts can be categorized into discrete prompts and continuous prompts. Discrete prompts use concrete word tokens to prompt pre-trained models. Compared to discrete prompts, continuous prompts (also called soft prompts) relax the token embedding space to be continuous. Hence, continuous prompts are differentiable and parameterized by their weights. They can be automatically tuned conditioning on downstream tasks.

Continuous prompts have shown competitive performance in language (Li & Liang, 2021; Gu et al., 2021; Liu et al., 2021; 2023), vision (Jia et al., 2022; Bahng et al., 2022; Han et al., 2023) and multimodality (Zhou et al., 2022b; Shu et al., 2022; Ju et al., 2022; Wang et al., 2022). Existing continuous prompting methods use the prompt depth and context length to design continuous prompts. The underlying constraint is that the context length remains constant at different depths. If different layers have different levels of deviation from the optimal weights for downstream tasks, the constraint might be detrimental to the performance.

Recent works (Lee et al., 2022; Chiatti et al., 2023; Panigrahi et al., 2023) have found that some layers of pre-trained models, depending on the distribution shift, are close to the optimal for downstream tasks. Fine-tuning layers that are far away from the optimal weights can achieve better performance than training all the layers uniformly. For prompting methods, we postulate that the layers far away from the optimal weights require longer context length while the layers close to the optimal weights demand shorter context length or even no context token. Hence, we seek to remove the constraint in the existing continuous prompting methods that require the same context length at different depths.

To this end, we propose a method dubbed **ada**ptive **p**rompt **t**uning (Adapt) that automatically determines context lengths at various depths. Adapt uses a time-dependent binary mask to dynamically control context lengths. The variation of the binary mask depends on the importance of context tokens. The least important context token is constantly removed until the budget (a hyperparameter to control the total context length) is reached. We test the performance of Adapt on various downstream task. Adapt outperforms baseline methods by a large margin as shown in Figure 1. To our best knowledge, this is the first work to prune prompts for achieving heterogeneous context lengths.

The main contribution of adapt is summarized as follows:

- We propose a method that removes the constraint in the existing continuous prompting methods that context lengths remain constant through the entire prompt depth. Adapt encourages a more flexible design for prompting methods.

- Context lengths are automatically determined in a non-parametric manner: prompts are initialized with the maximum context length and then iteratively pruned based on the importance score of context tokens. We use saliency criteria to characterize the importance of inserted context tokens. Pruning can effectively reduce the computational overhead with the minimal performance drop.

- Context lengths can vary based on the downstream datasets. We use a hyperparameter of total context lengths to ensure the complexity of Adapt on various datasets is approximately the same.

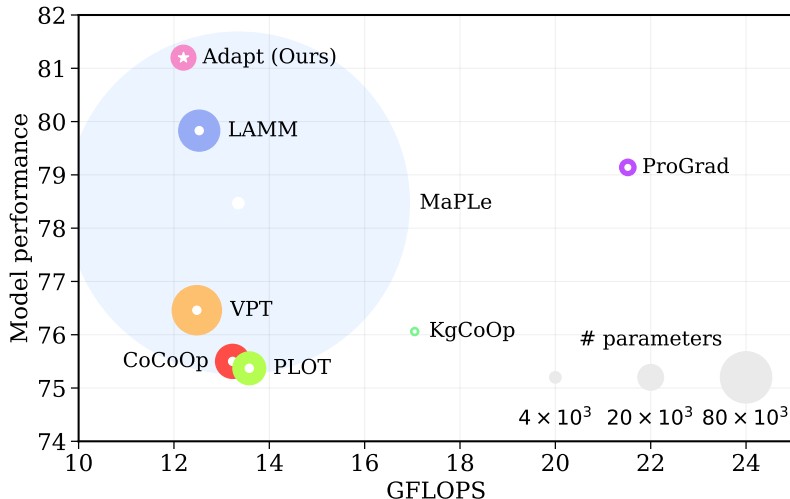

Figure 1: Average test accuracy over 11 datasets of different prompt tuning methods. Adapt surpasses state-of-the-art methods. We use Snip to compute importance scores and $\mathcal{T}_{\text{target}} = 32$.

## 2 RELATED WORK

**Prompt Tuning** Prompt tuning (PT) uses continuous prompts to improve the performance of pre-trained models in diverse downstream tasks. CoOp (Zhou et al., 2022b) is the pioneering work to apply PT for vision-language models. PT has shown great potential in various areas including image

classification (Zhou et al., 2022b;a; Hirohashi et al., 2024), out-of-distribution detection (Miyai et al., 2024; Li et al., 2024), video understanding (Ju et al., 2022; Huang et al., 2023), object detection (Du et al., 2022; He et al., 2023), etc. Due to the good alignment of text and image representations of foundational vision-language models, there are emerging researches on applying those models such as CLIP (Radford et al., 2021) to vision-language tasks. VPT (Jia et al., 2022) proposes a paradigm of deep continuous prompting. PLOT (Chen et al., 2022a) applies the optimal transport theory to improve the alignment between visual features and prompts. ProGrad (Zhu et al., 2023) and KgCoOp (Yao et al., 2023) distill the prior knowledge from the pre-trained model to avoid forgetting issues (Li & Hoiem, 2017; Gou et al., 2021). MaPLe (Khattak et al., 2023) uses linear transformation layers to enhance the coupling between the text and image branches. LAMM (Gao et al., 2024) uses dynamic category embedding and hierarchical loss to achieve an appropriate label distribution.

**Network Pruning** Over-parametrization is a well-known property of deep neural networks. Network pruning removes unimportant model parameters to improve efficiency. It can be categorized into structured pruning and unstructured pruning. Unstructured pruning such as (Han et al., 2015) removes individual parameters while structured pruning such as (Liu et al., 2018) prunes models at a higher level (*e.g.* neurons, filters, and layers). A fundamental question in network pruning is to identify a saliency criterion to determine the importance of model parameters. Snip (Lee et al., 2018) is a classic way to characterize the importance and can lead to a very sparse network without sacrificing too much performance.

**Sparse Training** Sparse training decreases a portion of model parameters based on the pre-defined pruning strategy. SViTE (Chen et al., 2021) iteratively uses a prune-and-grow strategy to update the model sparsity. Specifically, the linear transformation layers to query, key, and value are pruned. Mask tuning Zheng et al. (2023) updates those layers in a differentiable manner, *i.e.*, masks controlling sparsity is updated by learning instead of pruning. DRSformer Chen et al. (2023) utilizes learnable selection operators for attention scores in lieu of linear transformation layers.

In both network pruning and sparse training, sparsity distribution is a key factor. Concentrated pruning on a neural layer can cause the disconnection issue in neural networks. Nevertheless, when pruning soft prompts, this issue is inherently solved. If the entire deep soft prompts for a layer of a pre-trained model are pruned (*i.e.* context length is 0), it indicates that the effective prompt depth decreases by 1. This scenario is equivalent to the case in manually designed deep prompting methods where the depth of neural network layers is larger than the prompt depth.

## 3 ADAPTIVE PROMPT TUNING

We examine Adapt on the vision-language model CLIP (Radford et al., 2021). CLIP is pre-trained over 400 million image-text pairs. The pre-training process is in a contrastive learning fashion to promote the alignment between text and image representations. CLIP consists of an image encoder and a text encoder. The prediction is done by matching the text and image representations.

### 3.1 REVISITING CLIP

Given an input image $\mathbf{I} \in \mathbb{R}^{H \times W \times 3}$, the image encoder splits it into fixed-size patches that are projected into patch embeddings $\mathbf{x} \in \mathbb{R}^{(N_i-1) \times d_i}$ (Dosovitskiy, 2020). A learnable classification token embedding $\mathbf{c}_i^{(0)}$ is prepended to the patch embeddings. The concatenated sequence of embeddings is passed to $\ell$ transformer blocks:

$$[\mathbf{c}_i^{(j)}, \mathbf{E}_i^{(j)}] = f^{(j)}([\mathbf{c}_i^{(j-1)}, \mathbf{E}_i^{(j-1)}]) , \qquad (1)$$

where $j \in \mathbb{N}^+, 1 \leq j \leq \ell$, $f^{(j)}$ is the $j$-th transformer block of the image encoder. $\mathbf{E}_i^{(0)} = \mathbf{x}$. In the head of the image encoder, a linear transformation layer $\pi_i : \mathbb{R}^{d_i} \to \mathbb{R}^d$ transforms the classification token embedding in the image branch to the image representation $\mathbf{f}$.

A text prompt is fed to the text encoder to obtain the text embedding $\mathbf{E}_t = [\mathbf{w}^1, \mathbf{w}^2, \dots, \mathbf{w}^{N_t}] \in \mathbb{R}^{N_t \times d_t}$. The text embedding contains the classification token embedding as the first token embedding. The text embedding is passed to $\ell$ transformer blocks:

$$\mathbf{E}_t^{(j)} = g^{(j)}(\mathbf{E}_t^{(j-1)}) \,, \tag{2}$$

where $g^{(j)}$ is the $j$-th transformer block of the text encoder. In the head of the text encoder, a linear transformation layer $\pi_t : \mathbb{R}^{d_t} \to \mathbb{R}^d$ transforms the classification token embedding in the text branch to the text representation $\mathbf{g}$.

The prediction for the input image $\mathbf{I}$ is computed by the cosine similarity between the text embedding and the image embedding:

$$p(y = i | \mathbf{x}) = \frac{\exp(\cos(\mathbf{f}_i, \mathbf{g})/\tau)}{\sum_{j=1}^{K} \exp(\cos(\mathbf{f}_j, \mathbf{g})/\tau)} \,. \tag{3}$$

Here $\tau$ is the temperature parameter, $K$ is the total number of classes.

### 3.2 TIME-DEPENDENT PROMPT

Figure 2 (a) showcases the traditional shallow and deep prompts for vision language models. Figure 2 (b)-(d) show the proposed Adapt method. In the fine-tuning process of the pre-trained model, Adapt maximizes the likelihood of the correct label $y$:

$$\max_{\mathbf{P} \odot \mathcal{M}(t)} \mathbb{P}_{\mathbf{P} \odot \mathcal{M}(t), \boldsymbol{\theta}}(y | \mathbf{x}, \mathbf{P} \odot \mathcal{M}(t), \boldsymbol{\theta}) \,, \tag{4}$$

where $\boldsymbol{\theta}$ is the weight of the pre-trained model that is frozen during the fine-tuning process. $\mathbf{P} \in \mathbb{R}^{\ell \times \xi \times d}$ is the inserted continuous prompt. $\xi$ is the maximum context length at various depths. $\mathcal{M}(t) \in \{0, 1\}^{\ell \times \xi}$ is a time-dependent binary mask. We use $\odot$ to denote a modified Hadamard operation $\mathbf{M} = \mathbf{P} \odot \mathcal{M}(t)$, where $M_{ijk} = P_{ijk} \mathcal{M}(t)_{ij}, 1 \leq i \leq \ell, 1 \leq j \leq \xi, 1 \leq k \leq d$. For the vision-language model, there are two sets of independent prompts and binary masks. The optimation objective is over $\mathbf{P}_f \odot \mathcal{M}_f(t)$ and $\mathbf{P}_g \odot \mathcal{M}_g(t)$. $\mathbf{P}_f$ and $\mathbf{P}_g$ are prompts for image and text branches. $\mathcal{M}_f(t)$ and $\mathcal{M}_g(t)$ are binary masks for image and text branches.

We describe the optimization process of Adapt for the vision-language model as:

$$\underset{\mathbf{P}_f, \mathcal{M}_f(t), \mathbf{P}_g, \mathcal{M}_g(t)}{\mathrm{argmin}} \quad \frac{1}{|\mathcal{D}|} \sum_{\mathbf{x}, y \in \mathcal{D}} \mathcal{L}(\mathbf{x}, y | \mathbf{P}_f, \mathcal{M}_f(t), \mathbf{P}_g, \mathbf{M}_g(t), \boldsymbol{\theta}) \,,$$

$$\text{s.t.} \quad \sum_{i=1}^{\ell_f} \sum_{j=1}^{\xi_f} \mathcal{M}_f(t)_{ij} + \sum_{i=1}^{\ell_g} \sum_{i=1}^{\xi_g} \mathcal{M}_g(t)_{ij} \leq \mathcal{T}_{\text{target}} \,, \tag{5}$$

where the hyperparameter $\mathcal{T}_{\text{target}}$ is the target total context length. It determines the complexity of the Adapt method. For the brevity, we do not explicitly mention $\mathcal{M}_f(t)$ for the image branch and $\mathcal{M}_g(t)$ for the text branch. Instead we use $\mathcal{M}(t)$ as it can be applied to both the image and text branch. $\mathcal{M}(t)$ is initialized to be $\mathcal{M}(0) = \mathbf{1}_{\ell \times \xi}$. At each iteration, we identify which token to prune and set the corresponding binary mask to be 0, *i.e.* $\mathcal{M}(t)_{ij} = 0$. The total context length continuously decreases until $\mathcal{T}_{\text{target}}$ is reached. We use $\mathcal{T}_{\text{target}} \ll \ell \times \xi$ to ensure the efficiency of the Adapt method.

In the pruning process, which context token to prune, *i.e.* finding $i, j$ and set $\mathcal{M}(t)_{ij} = 0$, is determined by the importance of corresponding context tokens as shown in Figure 2 (d). We borrow the saliency criterion widely used in the unstructured network pruning literature to measure the importance of context tokens. Specifically, we use Snip (Lee et al., 2018), gradient norm and $l_2$-norm to compute the importance scores $S_c$ (also called saliency scores) for characterizing the importance. For the $t$-th context token ($t \in \mathbb{N}^+, 1 \leq t \leq \xi$) at the depth $l$ ($l \in \mathbb{N}^+, 1 \leq l \leq \ell$), the importance score computed by these three metrics is:

$$\text{Snip: } S_c = \left| \frac{\partial \mathcal{L}}{\partial \mathbf{P}_{lt}} \odot \mathbf{P}_{lt} \right|, \quad \text{gradient norm: } S_c = \left| \frac{\partial \mathcal{L}}{\partial \mathbf{P}_{lt}} \right|, \quad l_2\text{-norm: } S_c = |\mathbf{P}_{lt}| \,. \tag{6}$$

$\mathcal{M}(t)$ controls the context length for each transformer block as shown in Figure 2 (b). $\mathcal{M}(t) \odot \mathbf{P}$ is the continuous prompt inserted to the pre-trained model. There is no constraint for context lengths at various depths. Hence, the added prompt $\mathbf{P} \odot \mathcal{M}(t)$ can be heterogeneous.

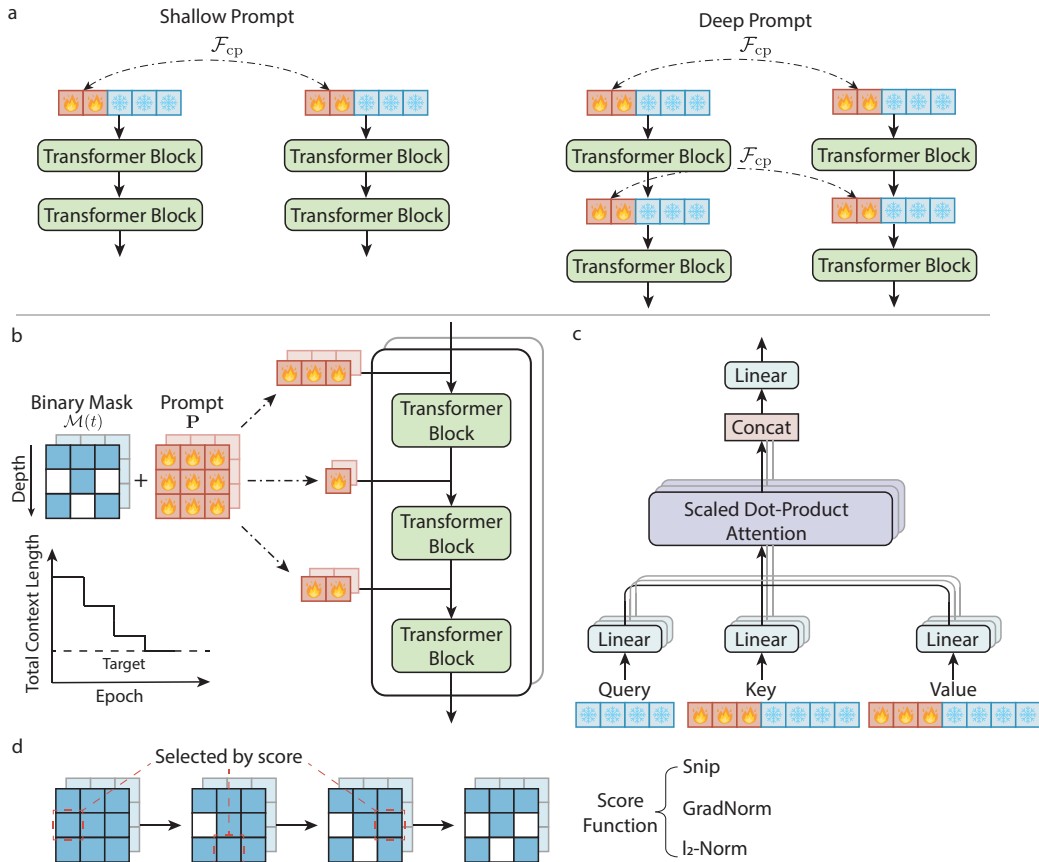

Figure 2: (a) The architecture paradigm of existing shallow and deep prompting methods. The former inserts prompts only into the inputs to the image and text encoders. The latter constantly replaces the inserted prompts from the last transformer block with newly inserted prompts for the current transformer block. Some works use a coupling function $\mathcal{F}_{cp}$ to bridge the text branch to the image branch. (b) The proposed Adapt method encourages the pre-trained model to insert prompts with different context lengths. We use two binary masks $\mathcal{M}_f(t)$ and $\mathcal{M}_g(t)$ to adaptively control context lengths. Context lengths constantly change until the target $\mathcal{T}_{\text{target}}$ is reached. Context lengths for these two branches at the same depth can be different. (c) In the multi-head attention, we insert continuous prompts for key and value computation. The backbone model is frozen during the fine-tuning process. Only continuous prompts are differentiable. (d) The selection of context tokens to be pruned is based on the importance scores. We use the saliency criterion in the unstructured pruning to compute scores.

### 3.3 PROMPT TUNING

Owing to $\mathcal{M}(t)$, the context length $\xi_l$ varies during the fine-tuning process. Unlike the existing deep prompting methods for the vision-language models that insert continuous prompts in the computation of key, value and query, Adapt inserts continuous prompts only for query and value in the self-attention (Vaswani et al., 2017) as shown in Figure 2 (c). Given an input $\mathbf{x}$ for a transformer block, the self-attention with inserted prompts in Adapt is computed by:

$$\mathbf{Q} = f_q(\mathbf{x}) \in \mathbb{R}^{\xi_{\text{org}} \times d}, \ \mathbf{K} = f_k([\mathbf{P}, \mathbf{x}]) \in \mathbb{R}^{(\xi_{\text{org}}+\xi) \times d}, \ \mathbf{V} = f_v([\mathbf{P}, \mathbf{x}]) \in \mathbb{R}^{(\xi_{\text{org}}+\xi) \times d} . \quad (7)$$

$$\text{Self-Attention} = \text{Softmax}(\frac{\mathbf{Q}\mathbf{K}^T}{\sqrt{d}})\mathbf{V} . \quad (8)$$

Here $f_q$, $f_k$ and $f_v$ are linear transformation functions for the query, key and value. $\xi_{\text{org}}$ is the sequence length of the input without the inserted prompt. During the fine-tuning process, the pre-trained model is frozen. Only inserted continuous prompts are optimized.

The proposed Adapt method for vision-language models is summarized in Algorithm 1. In the pruning step, the ranking is done based on scores in the image and text branches. The context tokens with the lowest score will be removed. The total context length in the text branch can be different from that in the image branch. For the same branch, context lengths might vary at different depths. Hence, compared to the manually designed continuous prompt, Adapt can have highly heterogeneous context lengths. Besides, using the saliency criterion enables varying context lengths without additional trainable parameters.

When the total context lengths of text and image branches reach $\mathcal{T}_{\text{target}}$, we do not remove context tokens. The accumulation period $n_k$ determines the number of accumulated steps to compute the score. The pruning rate $r_p$ dictates the number of removed tokens per pruning step.

---

**Algorithm 1** Adapt for vision-language models.

---

1: **Input**: A pre-trained vision-language model, prompt depth $\ell_f$ for the image encoder and $\ell_g$ for the text encoder, maximum context length $\xi_f$ for the image encoder and $\xi_g$ for the text encoder, target $\mathcal{T}_{\text{target}}$, accumulation period $n_k$ and pruning rate $r_p$.

2: Create a randomly initialized prompt $\mathbf{P}_f \in \mathbb{R}^{\ell_f \times \xi_f \times d}$ for the image branch and $\mathbf{P}_g \in \mathbb{R}^{\ell_g \times \xi_g \times d}$ for the text branch, and a binary mask $\mathcal{M}_f(0) = \mathbf{1}_{\ell_f \times \xi_f}$ for the image branch and $\mathcal{M}_g(0) = \mathbf{1}_{\ell_g \times \xi_g}$ for the text branch.

3: Initialize accumulated score $\mathbf{S}_f = \mathbf{0}_{\ell_f \times \xi_f}$ and $\mathbf{S}_g = \mathbf{0}_{\ell_g \times \xi_g}$.

4: **for** $t = 1, \ldots, n_t$ **do**  ▷ Loop through $n_t$ iterations

5:  Insert the prompt $\mathbf{P}_f \odot \mathcal{M}_f(t)$ for the image branch and $\mathbf{P}_g \odot \mathcal{M}_g(t)$ for the text branch of the pre-trained model as shown in Equation 7 and 8.

6:  Perform forward and backward propagation to update $\mathbf{P}_f$ and $\mathbf{P}_g$.

7:  **if** $\sum_{i=1}^{\ell_f} \sum_{j=1}^{\xi_f} \mathcal{M}_f(t)_{ij} + \sum_{i=1}^{\ell_g} \sum_{j=1}^{\xi_g} \mathcal{M}_g(t)_{ij} > \mathcal{T}_{\text{target}}$ **then**  ▷ When $\mathcal{T}_{\text{target}}$ is not reached, prune context tokens

8:   Compute scores $\Delta\mathbf{S}_f \in \mathbb{R}^{\ell_f \times \xi_f}$ and $\Delta\mathbf{S}_g \in \mathbb{R}^{\ell_g \times \xi_g}$ according to Equation 6.

9:   Update accumulated scores $\mathbf{S}_f$ by $\mathbf{S}_f = \mathbf{S}_f + \Delta\mathbf{S}_f$ and $\mathbf{S}_g$ by $\mathbf{S}_g = \mathbf{S}_g + \Delta\mathbf{S}_g$.

10:   **if** $t == an_k, a \in \mathbb{N}^+$ **then**

11:    **for** Prune step $= 1, \ldots, r_p$ **do**

12:     $(k_{\min}, i_{\min}, j_{\min}) = \operatorname{argmin}_{k,i,j}\{[\mathbf{S}_k]_{ij} | \mathcal{M}_k(t)_{ij} == 1\}$.  ▷ Find valid context tokens with the minimal score

13:     $\mathcal{M}_{k_{\min}}(t)_{i_{\min}j_{\min}} = 0$.  ▷ Prune context token

14:    **end for**

15:    Reset accumulated score $\mathbf{S}_f = \mathbf{0}_{\ell_f \times \xi_f}$ and $\mathbf{S}_g = \mathbf{0}_{\ell_g \times \xi_g}$.

16:   **end if**

17:  **end if**

18: **end for**

---

## 4 EXPERIMENTS AND RESULTS

### 4.1 EXPERIMENTS

**Datasets**  We examine the proposed Adapt method over 11 datasets: Caltech101 (Fei-Fei et al., 2004) and ImageNet (Deng et al., 2009) for the generic object recognition, DescribableTextures (Cimpoi et al., 2014) for the texture recognition, EuroSAT (Helber et al., 2019) for the satellite image recognition, FGVCAircraft (Maji et al., 2013), Food101 (Bossard et al., 2014), OxfordFlowers (Nilsback & Zisserman, 2008), OxfordPets (Parkhi et al., 2012), and StanfordCars (Krause et al., 2013) for the fine-grained image recognition, UCF101 (Soomro et al., 2012) for the action recognition, and SUN397 (Xiao et al., 2010) for the scene recognition. We follow the few-shot learning setting in CoOp (Zhou et al., 2022b). The number of shots is 16. For each dataset, the result is averaged over 3 runs. A detailed description of 11 datasets can be found in the Appendix.

**Baselines** We compare the proposed method with CoCoOp (Zhou et al., 2022a), VPT (Jia et al., 2022), PLOT (Chen et al., 2022a), MaPLe (Khattak et al., 2023), ProGrad (Zhu et al., 2023), Kg-CoOp (Yao et al., 2023) and LAMM (Gao et al., 2024). The original implementation of PLOT uses ResNet (He et al., 2016) in the image encoder. For a fair comparison, we replace ResNet with ViT in the image encoder. CoCoOp, PLOT, ProGrad, KgCoOp and LAMM use shallow prompts while VPT and MaPLe use deep prompts. CoCoOp adds continuous prompts conditioned on the input Image. PLOT uses the optimal transport theory to align the vision and text modalities. ProGrad uses gradient-aligned knowledge distillation to alleviate the forgetting issue in the fine-tuning process. KgCoOp uses the prior knowledge from the hand-crafted prompt in the knowledge distillation. LAMM replaces the category tokens with trainable vectors and utilizes the hierarchical loss to preserve the generalization ability of the pre-trained model. VPT proposes a deep prompting method. MaPLe uses a coupling function to bridge the image branch and text branch.

**Implementation Details** We use the pretrained ViT-B/16 CLIP model (Radford et al., 2021) in this work. The number of minibatches used for computing the score is $n_k = 80$. In each pruning step, the number of pruned context tokens is $r_p = 4$. The batch size is 4. The learning rate is $2.5 \times 10^{-3}$. The total number of training epochs is 100. The test accuracy is obtained using the model weights at the epoch of 100. We use the stochastic gradient descent (SGD) to optimize the inserted prompts. Experiments are conducted using a single NVIDIA A40 GPU. Reported results on 11 datasets are averaged over 3 runs.

Table 1: Test accuracy comparison on various downstream tasks in the few-shot learning setting. We report both the total number of trainable parameters and the percentage of those parameters on top of the pre-trained CLIP. Adapt uses Snip to compute scores of context tokens. Adapt (Adaptive $\mathcal{T}_{\text{target}}$) uses the validation set to automatically select $\mathcal{T}_{\text{target}}$. Details are described in Appendix A.9.

| Method | # Trainable Params | GFLOPS | Caltech101 | DTD | EuroSAT | Aircraft | Food101 |
|---|---|---|---|---|---|---|---|
| ZS CLIP | 0 (0%) | 12.08 | 87.20 | 42.34 | 37.57 | 17.29 | 77.30 |
| CoCoOp | 35,360 (0.028%) | 13.23 | 95.10 | 63.63 | 74.10 | 33.67 | 87.37 |
| VPT | 73,728 (0.059%) | 12.48 | 94.83 | 67.30 | 86.23 | 33.90 | 87.03 |
| PLOT | 32,768 (0.026%) | 13.58 | 93.70 | 70.90 | 84.03 | 34.93 | 78.13 |
| MaPLe | 3.56 M (2.860%) | 13.35 | 95.10 | 67.27 | 86.40 | 37.07 | **87.43** |
| ProGrad | 8,192 (0.007%) | 21.52 | 95.63 | 66.27 | 82.03 | 41.30 | 86.70 |
| KgCoOp | 2,048 (0.002%) | 17.05 | 95.07 | 67.00 | 72.80 | 34.17 | 87.07 |
| LAMM | 51,200 (0.041%) | 13.23 | 95.67 | 70.43 | 84.43 | 41.27 | 87.10 |
| Adapt ($\mathcal{T}_{\text{target}} = 128$) | 82,227 (0.066%) | 12.53 | 95.63 | **72.03** | **92.53** | **50.93** | 83.47 |
| Adapt ($\mathcal{T}_{\text{target}} = 32$) | 18,781 (0.015%) | 12.20 | **96.10** | 69.93 | 90.13 | 48.73 | 84.30 |
| Adapt (Adaptive $\mathcal{T}_{\text{target}}$) | - | - | 96.17 | 72.17 | 92.60 | 52.07 | 87.03 |
| Method | Flowers | Pets | Cars | Sun | UCF | ImageNet | Average |
| ZS CLIP | 66.18 | 85.79 | 55.63 | 58.55 | 61.45 | 58.20 | 58.86 |
| CoCoOp | 89.97 | 93.53 | 72.30 | 72.67 | 76.97 | 71.17 | 75.50 |
| VPT | 88.10 | 92.57 | 69.60 | 71.87 | 79.00 | 70.60 | 76.46 |
| PLOT | 97.27 | 88.20 | 68.10 | 69.40 | 72.23 | 72.17 | 75.37 |
| MaPLe | 94.27 | **93.63** | 74.87 | 74.73 | 80.37 | 72.03 | 78.47 |
| ProGrad | 95.33 | 93.10 | 81.23 | **75.13** | 81.60 | **72.27** | 79.14 |
| KgCoOp | 90.00 | 92.93 | 73.33 | 73.00 | 80.63 | 70.60 | 76.06 |
| LAMM | 95.93 | 93.53 | 82.87 | 73.27 | 81.60 | 72.03 | 79.83 |
| Adapt ($\mathcal{T}_{\text{target}} = 128$) | **97.97** | 91.07 | **86.17** | 73.67 | **84.40** | 70.83 | **81.70** |
| Adapt ($\mathcal{T}_{\text{target}} = 32$) | 97.63 | 90.83 | 85.07 | 74.53 | 84.03 | 71.93 | 81.20 |
| Adapt (Adaptive $\mathcal{T}_{\text{target}}$) | 98.40 | 92.47 | 86.70 | 75.33 | 84.40 | 72.07 | 82.67 |

## 4.2 RESULTS

The effectiveness of the Adapt method is examined using the few-shot learning setting. We summarize the experimental results in Table 1. We use $\ell_f = \ell_g = 12$ and $\xi_f = \xi_g = 16$. Overall, the Adapt method exhibits superior performance compared to baseline methods. Adapt ($\mathcal{T}_{\text{target}} = 128$) achieves the overall gain from 79.83% to 81.70% on the average of 11 datasets. The large performance gain is 6.13% on the EuroSAT and 9.63% on the Aircraft dataset. Adapt relies merely on inserting continuous prompts with different lengths. Baseline methods except for VPT (Jia et al., 2022) replies on additional assistance such as knowledge distillation (Hinton et al., 2015). Hence, Adapt has the second lowest GLOPS. PLOT (Chen et al., 2022a) uses an iteration algorithm to com-

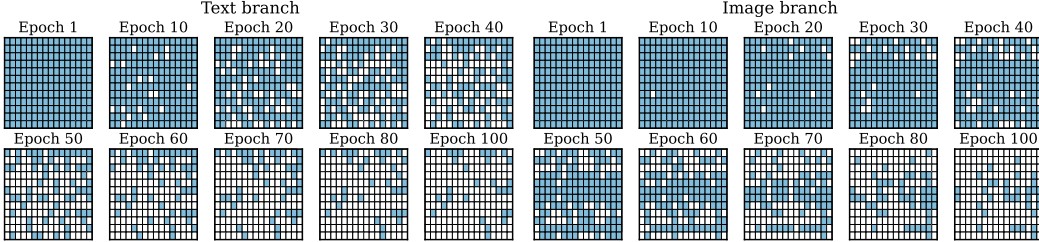

Figure 3: Pruning process of binary masks in the text and image branches on the EuroSAT dataset. $\mathcal{T}_{\text{target}} = 64$. The warmup epoch is 5, so there is no pruning of context tokens in the first 5 epochs. Given $\mathcal{M}(t)$ matrices, the row dimension corresponds to the prompt depth and the column dimension corresponds to the maximum context length.

Table 2: Performance comparison of different $\mathcal{T}_{\text{target}}$. We use Snip to compute the score of each context token.

| Method | # Trainable Params | GFLOPS | Caltech101 | DTD | EuroSAT | Aircraft | Food101 |
|---|---|---|---|---|---|---|---|
| Adapt ($\mathcal{T}_{\text{target}} = 512$) | 0.36 M (0.2889%) | 12.98 | 96.03 | 72.07 | 92.30 | 52.03 | 83.37 |
| Adapt ($\mathcal{T}_{\text{target}} = 256$) | 0.17 M (0.1380%) | 12.73 | 95.53 | 72.60 | 92.23 | 51.37 | 83.43 |
| Adapt ($\mathcal{T}_{\text{target}} = 128$) | 82,227 (0.066%) | 12.53 | 95.63 | 72.03 | 92.53 | 50.93 | 83.47 |
| Adapt ($\mathcal{T}_{\text{target}} = 64$) | 39,168 (0.032%) | 12.32 | 95.93 | 70.60 | 91.87 | 49.17 | 83.83 |
| Adapt ($\mathcal{T}_{\text{target}} = 32$) | 18,781 (0.015%) | 12.20 | 96.10 | 69.93 | 90.13 | 48.73 | 84.30 |
| Adapt ($\mathcal{T}_{\text{target}} = 16$) | 8,986 (0.007%) | 12.11 | 95.53 | 67.17 | 91.00 | 40.77 | 84.93 |

| Method | Flowers | Pets | Cars | Sun | UCF | ImageNet | Average |
|---|---|---|---|---|---|---|---|
| Adapt ($\mathcal{T}_{\text{target}} = 512$) | 98.40 | 89.93 | 86.43 | 73.30 | 84.23 | 70.57 | 81.70 |
| Adapt ($\mathcal{T}_{\text{target}} = 256$) | 98.13 | 90.20 | 86.03 | 73.73 | 84.10 | 70.77 | 81.65 |
| Adapt ($\mathcal{T}_{\text{target}} = 128$) | 97.97 | 91.07 | 86.17 | 73.67 | 84.40 | 70.83 | 81.70 |
| Adapt ($\mathcal{T}_{\text{target}} = 64$) | 98.10 | 90.57 | 85.13 | 73.83 | 83.30 | 71.03 | 81.21 |
| Adapt ($\mathcal{T}_{\text{target}} = 32$) | 97.63 | 90.83 | 85.07 | 74.53 | 84.03 | 71.93 | 81.20 |
| Adapt ($\mathcal{T}_{\text{target}} = 16$) | 97.57 | 90.43 | 83.30 | 75.33 | 84.07 | 71.43 | 80.14 |

pute the optimal transport plan, which is ignored in the FLOPS calculation. Details regarding the iteration algorithm are reported in (Cuturi, 2013). The computational costs for the Adapt method are reported based on binary masks at the final epoch. All continuous prompting methods except for MaPLe (Khattak et al., 2023) have trainable parameters accounting to less than 0.1% of all ViT-Base parameters.

Adapt inserts continuous prompts only for the key and value computations while the prevalent deep prompting methods for vision-language models insert continuous prompts for query, key and value computations. Using the same context length, this approach can effectively decrease FLOPs. Besides, the continuous prompts are not added to the query, which does not change the context length after the attention computation. Therefore, prompts with heterogeneous context lengths can be added to the pre-trained model.

A typical pruning process for $\mathcal{M}_f(t)$ and $\mathcal{M}_g(t)$ is shown in Figure 3. The binary mask at the epoch of 1 is the same as the initialized mask due to the warmup process. Context lengths in the text and image branches are highly heterogeneous: context lengths at the image branch are different from those in the text branch. Within the same branch, context lengths vary at various depths. When $\mathcal{T}_{\text{target}}$ is reached, context lengths stay constant. We use $\mathcal{T}_{\text{target}} \ll \ell \times \xi$, $\mathcal{M}_f(t)$ and $\mathcal{M}_g(t)$ are sparse matrices after training. The pruning processes for all 11 datasets are shown in Appendix Figure 8. We track the variation of context lengths as a function of the number of training epochs. The result is shown in Appendix Figure 4.

When allowing dataset-dependent $\mathcal{T}_{\text{target}}$ denoted as Adapt (Adaptive $\mathcal{T}_{\text{target}}$), the average performance can be boosted to 82.67%. Adapt (Adaptive $\mathcal{T}_{\text{target}}$) uses the validation dataset to select $\mathcal{T}_{\text{target}}$. Details regarding Adapt (Adaptive $\mathcal{T}_{\text{target}}$) are described in Appendix A.9.

### 4.3 ABLATION STUDY

**Target total context length** $\mathcal{T}_{\text{target}}$ $\quad$ $\mathcal{T}_{\text{target}}$ is associated with the complexity of inserted prompts. Table 2 reports the performance of using different $\mathcal{T}_{\text{target}}$. The hidden dimension in the image encoder is not equal to that in the text encoder, *i.e.* $d_i \neq d_t$, for CLIP, the same $\mathcal{T}_{\text{target}}$ can lead to a different number of trainable parameters. The number of trainable parameters is averaged over 11 datasets.

When $\mathcal{T}_{\text{target}}$ is decreased from 128 to 64, the number of trainable parameters decreases by 52.37%, the performance drop is only 0.60%. Upon further reducing $\mathcal{T}_{\text{target}}$ to 32, the total number of parameters decreases by 77.16%, and the performance drop is 0.61%. The relatively small drop in the performance justifies the pruning of context tokens, *i.e.* update of $\mathcal{M}(t)$. When $\mathcal{T}_{\text{target}}$ is decreased to 16, there is a pronounced performance drop, especially on Aircraft dataset where the performance drop is 19.95%. The zero-shot transfer performance of CLIP is relatively poor on the Aircraft dataset. Adapt improves the performance from 17.29% to 50.93%. A larger complexity of inserted prompts is beneficial to the performance on this dataset. The performance on different $\mathcal{T}_{\text{target}}$ indicates that when the complexity is large enough, pruning of prompts, similar to network pruning, can improve the efficiency without negatively affecting the performance too much.

When increasing $\mathcal{T}_{\text{target}}$ to be larger than 128, there is no increase in the average test accuracy. Some datasets prefer a large complexity. For example, on Aircraft dataset, there is consistent performance gain when increasing $\mathcal{T}_{\text{target}}$.

**Score computation** $\quad$ We examine the effect of three different scoring functions: Snip (Lee et al., 2018), gradient norm, and $l_2$-norm. Table 3 shows the performance comparison. Snip considers both the gradient and magnitude of the prompt parameters. Snip has the best performance. Overall, there is no remarkable difference among score functions.

Table 3: Performance comparison using different score functions: Snip, gradient norm and $l_2$-norm. We use $\mathcal{T}_{\text{target}} = 128$. Owing to the difference between $d_t$ and $d_i$, the same $\mathcal{T}_{\text{target}}$ can lead to a different number of trainable parameters.

| Method | # Trainable Params | GFLOPS | Caltech101 | DTD | EuroSAT | Aircraft | Food101 |
|---|---|---|---|---|---|---|---|
| Adapt (Snip) | 82,227 (0.066%) | 12.53 | 95.63 | 72.03 | 92.53 | 50.93 | 83.47 |
| Adapt (Gradient Norm) | 84,044 (0.068%) | 12.53 | 95.63 | 72.07 | 91.83 | 50.93 | 83.63 |
| Adapt ($l_2$-Norm) | 82,764 (0.067%) | 12.53 | 95.57 | 70.97 | 91.47 | 51.17 | 83.77 |
| Method | Flowers | Pets | Cars | Sun | UCF | ImageNet | Average |
| Adapt (Snip) | 97.97 | 91.07 | 86.17 | 73.67 | 84.40 | 70.83 | 81.70 |
| Adapt (Gradient Norm) | 98.20 | 90.30 | 86.07 | 73.93 | 84.40 | 69.83 | 81.53 |
| Adapt ($l_2$-norm) | 98.17 | 90.33 | 85.83 | 74.03 | 84.37 | 70.23 | 81.45 |

## 5 DISCUSSION

When tailoring a pre-trained model to various downstream tasks, the model can underperform due to the distribution shift (DS) (Taori et al., 2020; Fang et al., 2020; Wiles et al., 2021; Xiao et al., 2024). When examining the model on a more granular level, a question arises "*is the inferior performance caused by the deviation from the optimal for all layers or a subset of layers*". Surgical fine-tuning (Lee et al., 2022) answers this question by categorizing DS into four categories: input-level shift, feature-level shift, output-level shift, and natural shift. Depending on the DS type, fine-tuning the selective part of the pre-trained model achieves a performance comparable to or better than training all layers. This result indicates that not all layers are at the same level of deviating from the optimal. For example, when DS is the input-level shift, only the first few layers are deviating away from the optimal. Training those layers while keeping the remaining layers frozen achieves favorable performance.

In the PT, the entire pre-trained model is frozen. Given the fact that some layers, depending on the DS type, might already be close to the optimal, there is no need to insert continuous prompts for those layers. Prompts can be inserted into layers that are deviating from the optimal. If we consider this strategy in a more granular way, context lengths for different layers can vary depending on

the level of deviating from the optimal. This leads to heterogeneous context lengths which are challenging for the manually designed prompting methods.

The proposed Adapt method achieves the automatic design of heterogeneous prompts. There is no constraint for context lengths at various depths to be the same, nor for context lengths to be the same for different branches. The results on 11 datasets indicate that context lengths can be highly heterogeneous as shown in Appendix Figure 8. The automation is achieved by iteratively pruning unimportant context tokens. By setting $\mathcal{T}_{\text{target}} \ll \ell \times \xi$, the pruning greatly reduces the computational overheads. We empirically find the performance of pruned prompts $\mathbf{P} \odot \mathcal{M}(t)$ is comparable to that of training prompts $\mathbf{P}$ without pruning from scratch as indicated in the Appendix Table 4. At the same time, the total number of trainable parameters is decreased by $67\%$. In the network pruning, pruning concentrated on one layer can cause the layer collapse issue (Lee et al., 2019; Hayou et al., 2020). Pruning prompts, however, can have the minimal context length in one layer without affecting the functionality of prompts for this layer.

By using $\mathcal{M}(t)$ conditioning on downstream datasets, Adapt adaptively changes for different datasets. Compared to manually designed prompts, Adapt has a more flexible structure. It achieves a pronounced performance gain compared to baseline methods. We use $\mathcal{T}_{\text{target}}$ to ensure the complexity of Adapt is approximately the same over various datasets.

**Limitation** While Adapt achieves the best average performance on the considered downstream tasks, we acknowledge that when the number of shots decreases (e.g., less than 4 shots), Adapt may lose its advantages. However, most prompting methods assume at least 16-shots per category (Hirohashi et al., 2024), which is the regime where Adapt outperforms the competitive methods. We believe the limitation of Adapt in very few-shot settings may be addressed by considering some specialized methods that cater to few-shot prompting, such as (Hirohashi et al., 2024) explicitly designed for 1-shot setting.

## 6 CONCLUSION

We propose a continuous prompting method that adaptively changes during the fine-tuning process. Different from existing prompting methods that require homogeneous context lengths for various depths, our proposed method Adapt encourages heterogeneous context lengths. Adapt uses iterative pruning to remove unimportant context tokens, which greatly reduces the computational costs with nearly no performance drop. Extensive experiments over 11 datasets exhibit the strength of the Adapt method.

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

# A APPENDIX

## A.1 COMPARISON BETWEEN PRUNING AND UNPRUNING

Adapt uses $\mathbf{P} \odot \mathcal{M}(t)$ in the fine-tuning of the pre-trained model. We compare its performance with training $\mathbf{P}$ from scratch without using $\mathcal{M}(t)$, which is essentially the Adapt method with $\mathcal{T}_{\text{target}} \geq \ell \times \xi$. We summarize the performance variation in Table 4. $\mathbf{P} \odot \mathcal{M}(t)$ uses merely 33% parameters of $\mathbf{P}$ and achieves comparable performance.

Similar to the context length study, there is a relatively large performance drop on Aircraft dataset. This might be due to the large difference between the pre-trained datasets and fine-tuning datasets. More trainable parameters are needed to adapt the pre-trained models to the downstream task.

Table 4: Performance variation compared to unpruned continuous prompts. $\mathcal{T}_{\text{target}} = 128$ and Snip is used to compute scores. We use blue color to indicate the performance increase and red color to indicate the performance drop.

| Caltech101 | DTD | EuroSAT | Aircraft | Food101 | Flowers | Pets | Cars | Sun | UCF | ImageNet | Average |
|---|---|---|---|---|---|---|---|---|---|---|---|
| -0.4 | -0.04 | 0.20 | -1.14 | 0.17 | -0.43 | 1.14 | -0.53 | 0.40 | 0.40 | 0.2 | -0.003 |

## A.2 TIME-DEPENDENT CONTEXT LENGTH

Figure 4 shows the variation of context lengths during the fine-tuning process. When $\mathcal{T}_{\text{target}}$ is reached, the total context length for the image and text branches stays constant. We use the warmup strategy and hence the total context length does not change for the first few epochs. In the pruning process, we prune the context tokens in the text and image branches with the lowest score. The number of tokens removed per epoch can vary from the image branch to the text branch. Within the same branch, the context length can vary at various depths. Overall, Adapt encourages highly heterogeneous context lengths which can be difficult for the manually designed prompts. Adapt is able to automatically determine context lengths. The pruning process on all 11 datasets is visualized in Figure 8.

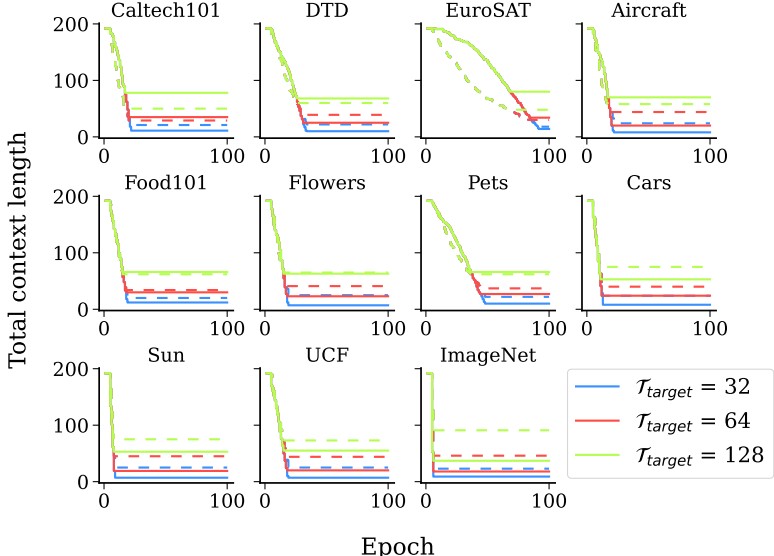

Figure 4: Variation of the total context length during the fine-tuning process for various downstream tasks. Solid lines are the total context length for the text branch while dashed lines correspond to the image branch.

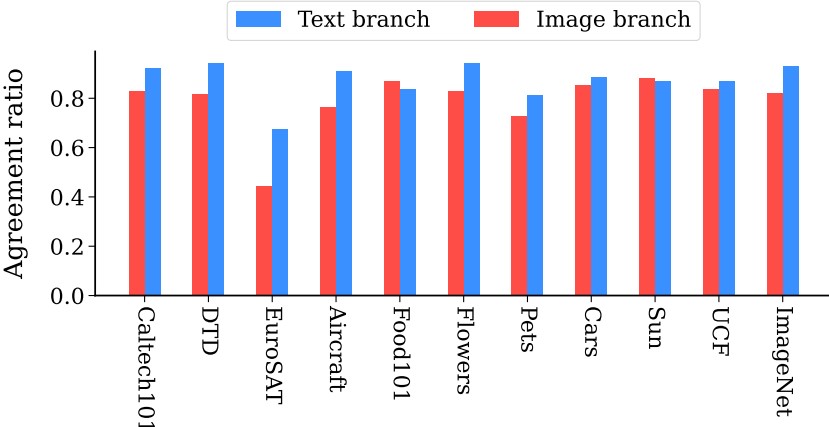

Figure 5: Agreement ratio of binary masks at the last epoch in the text and image branches for $r_p = 1$ and $r_p = 4$. The average agreement ratio for the text branch is 0.87 while that for the image branch is 0.79.

A.3 EFFECT OF PRUNING RATE

$r_p$ determines the rate of binary mask reaching $\mathcal{T}_{\text{target}}$. To study the effect of $r_p$ on the final binary mask, we use the agreement ratio to characterize the agreement of the binary mask between two different pruning rates:

$$\text{Agreement ratio} := \frac{1}{\ell \times \xi} \sum_{i=1}^{\ell} \sum_{j=1}^{\xi} \mathbb{I}\{\mathcal{M}_1(t)_{ij} = \mathcal{M}_2(t)_{ij}\} , \tag{9}$$

where $\mathbb{I}$ is the indicator function and $\mathbb{I}\{E\} = 1$ if the event $E$ happens, otherwise $\mathbb{I}\{E\} = 0$. $\mathcal{M}_1(t)$ and $\mathcal{M}_2(t)$ are binary matrices using two different pruning rates. The agreement ratio is in the range $[0, 1]$. A higher agreement ratio indicates two binary masks are closer.

Figure 5 shows the agreement ratio of the final binary masks on 11 datasets comparing $r_p = 1$ and $r_p = 4$. Overall, there is a high agreement ratio on all datasets. A high agreement ratio indicates the robustness of the Adapt method against $r_p$. Before $\mathcal{T}_{\text{target}}$ is reached, the prompt tuning by updating $\mathbf{P}$ and the pruning by updating $\mathcal{M}(t)$ happen simultaneously. In this early stage, $r_p = 1$ has a larger total context length compared to $r_p = 4$. Extra context tokens have a limited effect on the relative magnitude of the scores of context tokens.

A.4 TRAIN FINAL BINARY MASKS FROM SCRATCH

In network pruning, an empirical experience is that the sparse architectures produced by pruning are difficult to train from scratch. Lottery ticket hypothesis (LTH) (Frankle & Carbin, 2018) proposes the parameter reinitialization trick after each pruning step to identify the winning ticket. We use the binary masks at the final training epoch to determine context lengths. Using the fixed context lengths, we train continuous prompts from scratch. During the training process, the total context length remain constant.

Table 5 summarizes the accuracy difference between $\mathbf{P}$ and $\mathbf{P} \odot \mathcal{M}$ ($\mathcal{M}$ is fixed). The accuracy is comparable to the unpruned prompt using the same training epochs. At the same time, $\sum_{i=1}^{\ell} \sum_{j=1}^{\xi} \mathcal{M}_{ij} \ll |\ell \times \xi|$. Note that $\mathbf{P} \odot \mathcal{M}$, same as LTH, performs pruning at initialization while Adapt using $\mathbf{P} \odot \mathcal{M}(t)$ prunes prompts during training.

A.5 DATASETS FOR DOWNSTREAM TASKS

Table 6 shows statistics of 11 datasets used for the fine-tuning of the pre-trained CLIP model. These datasets covering a wide range of tasks are commonly used as benchmarks for vision-language models.

Table 5: Performance variation using fixed binary masks compared to prompting method without using masks (the highest total context lengths). We use blue color to indicate that using fixed binary masks has a better performance than adaptive prompts while red color to indicate the performance drop.

| Caltech101 | DTD | EuroSAT | Aircraft | Food101 | Flowers | Pets | Cars | Sun | UCF | ImageNet | Average |
|---|---|---|---|---|---|---|---|---|---|---|---|
| -0.03 | -0.67 | 0.27 | -0.24 | 0.87 | -0.77 | 1.37 | -0.97 | 1.03 | 0.50 | -0.83 | 0.05 |

Table 6: Details regarding 11 datasets used in experiments.

| Dataset | # Classes | Training size | Test size | Task |
|---|---|---|---|---|
| Caltech101 (Fei-Fei et al., 2004) | 100 | 4,128 | 2,465 | Generic object classification |
| ImageNet (Deng et al., 2009) | 1,000 | 1.28M | 50,000 | Generic object classification |
| DescribableTextures (Cimpoi et al., 2014) | 47 | 2,820 | 1692 | Texture classification |
| EuroSAT (Helber et al., 2019) | 10 | 13,500 | 8,100 | Satellite image classification |
| FGVCAircraft (Maji et al., 2013) | 100 | 3,334 | 3,333 | Fine-grained aircraft classification |
| Food101 (Bossard et al., 2014) | 101 | 50,500 | 30,300 | Fine-grained food classification |
| OxfordFlowers (Nilsback & Zisserman, 2008) | 102 | 4,093 | 2,463 | Fine-grained flower classification |
| OxfordPets (Parkhi et al., 2012) | 37 | 2,944 | 3,669 | Fine-grained pet classification |
| StanfordCars (Krause et al., 2013) | 196 | 6,509 | 8,041 | Fine-grained car classification |
| UCF101 (Soomro et al., 2012) | 101 | 7,639 | 3,783 | Action classification |
| SUN397 (Xiao et al., 2010) | 397 | 15,880 | 19,850 | Scene classification |

## A.6 PERFORMANCE COMPARISON

We compare the performance of the Adapt method with baseline methods. Table 7 shows the performance with standard deviation. Results are obtained over 3 different runs. There is no significant performance variation in the few-shot learning experiments of all prompting methods.

Table 7: Standard deviation of the performance of 16-shot learning on 11 datasets. The average performance is obtained over 3 runs.

| Method | Caltech101 | DTD | EuroSAT | Aircraft | Food101 | Flowers |
|---|---|---|---|---|---|---|
| CoCoOp | $95.10 \pm 0.08$ | $63.63 \pm 0.88$ | $74.10 \pm 0.57$ | $33.67 \pm 0.33$ | $87.37 \pm 0.12$ | $89.97 \pm 1.03$ |
| VPT | $94.83 \pm 0.42$ | $67.30 \pm 2.08$ | $86.23 \pm 0.79$ | $33.90 \pm 1.81$ | $87.03 \pm 0.22$ | $88.10 \pm 0.88$ |
| PLOT | $93.70 \pm 0.10$ | $70.90 \pm 0.54$ | $84.03 \pm 0.59$ | $34.93 \pm 1.05$ | $78.13 \pm 0.21$ | $97.27 \pm 0.12$ |
| MaPLe | $95.10 \pm 0.16$ | $67.27 \pm 0.61$ | $86.40 \pm 1.47$ | $37.07 \pm 0.25$ | $87.43 \pm 0.09$ | $94.27 \pm 0.13$ |
| ProGrad | $95.63 \pm 0.39$ | $66.27 \pm 0.73$ | $82.03 \pm 1.52$ | $41.30 \pm 0.49$ | $86.70 \pm 0.08$ | $95.33 \pm 0.38$ |
| KgCoOp | $95.07 \pm 0.31$ | $67.00 \pm 2.66$ | $72.80 \pm 1.81$ | $34.17 \pm 0.62$ | $87.07 \pm 0.38$ | $90.00 \pm 0.56$ |
| LAMM | $95.67 \pm 0.20$ | $70.43 \pm 0.62$ | $84.43 \pm 2.66$ | $41.27 \pm 0.43$ | $87.10 \pm 0.20$ | $95.93 \pm 0.13$ |
| Adapt ($\mathcal{T}_{target} = 128$) | $95.63 \pm 0.40$ | $72.03 \pm 1.66$ | $92.53 \pm 0.56$ | $50.93 \pm 1.30$ | $83.47 \pm 0.32$ | $97.97 \pm 0.11$ |
| Adapt ($\mathcal{T}_{target} = 32$) | $96.10 \pm 0.21$ | $69.93 \pm 1.81$ | $90.13 \pm 1.37$ | $48.73 \pm 1.01$ | $84.30 \pm 0.25$ | $97.63 \pm 0.22$ |

| Method | Pets | Cars | Sun | UCF | ImageNet |
|---|---|---|---|---|---|
| CoCoOp | $93.53 \pm 0.45$ | $72.30 \pm 0.54$ | $72.67 \pm 0.05$ | $76.97 \pm 0.85$ | $71.17 \pm 0.05$ |
| VPT | $92.57 \pm 0.42$ | $69.60 \pm 0.85$ | $71.87 \pm 0.48$ | $79.00 \pm 0.42$ | $70.60 \pm 0.22$ |
| PLOT | $88.20 \pm 0.51$ | $68.10 \pm 0.51$ | $69.40 \pm 0.16$ | $72.23 \pm 0.12$ | $72.17 \pm 0.10$ |
| MaPLe | $93.63 \pm 0.34$ | $74.87 \pm 0.68$ | $74.73 \pm 0.05$ | $80.37 \pm 0.78$ | $72.03 \pm 0.12$ |
| ProGrad | $93.10 \pm 0.37$ | $81.23 \pm 0.57$ | $75.13 \pm 0.25$ | $81.60 \pm 0.71$ | $72.27 \pm 0.05$ |
| KgCoOp | $92.93 \pm 0.78$ | $73.33 \pm 1.11$ | $73.00 \pm 0.49$ | $80.63 \pm 1.33$ | $70.60 \pm 0.66$ |
| LAMM | $93.53 \pm 0.20$ | $82.87 \pm 0.62$ | $73.27 \pm 0.31$ | $81.60 \pm 0.79$ | $72.03 \pm 0.10$ |
| Adapt ($\mathcal{T}_{target} = 128$) | $91.07 \pm 0.79$ | $86.17 \pm 0.09$ | $73.67 \pm 0.33$ | $84.40 \pm 0.40$ | $70.83 \pm 0.18$ |
| Adapt ($\mathcal{T}_{target} = 32$) | $90.83 \pm 0.43$ | $85.07 \pm 0.17$ | $74.53 \pm 0.13$ | $84.03 \pm 0.54$ | $71.93 \pm 0.36$ |

## A.7 PRUNING PROCESS OF BINARY MASKS

Figure 8 shows the pruning process of binary masks in the text and image branches for 11 datasets. The binary mask is initialized to be $\mathcal{M}(0) = \mathbf{1}_{\ell \times \xi}$. $\mathcal{M}(t)$ stays constant when $\mathcal{T}_{target}$ is reached. We use Snip to compute the score for each context tokens. The binary masks are highly heterogeneous: context lengths over prompt depth for different branches have a large variation. The heterogeneity feature exhibits the strength of the automatic design of context lengths compared to manually designed prompts which tend to be homogeneous.

When $\mathcal{T}_{\text{target}}$ is reached, $\mathbf{P} \odot \mathcal{M}(t)$ on some datasets have a context length of 1 at a certain depth. Even though the pruning concentrated in the prompts for this layer, the performance is not negatively affected. In the network pruning, however, the concentrated pruning can lead to the layer collapse issue (Lee et al., 2019; Hayou et al., 2020). Considering an extreme case, if the weight of an entire layer is pruned, the model will have a disconnection issue. When pruning the prompt, there is no such issue. This indicates that pruning prompts can lead to highly heterogeneous prompt lengths.

## A.8    ANALYSIS OF BINARY MASKS

We examine the total context lengths on the text and image branches using $\mathcal{T}_{\text{Target}} = 128$. Figure 6 shows the total context lengths on the text and image branches over 11 datasets. The context lengths at different depths are shown in Figure 8. (Tsao et al., 2023) finds that EuroSAT has a relatively large distribution shift (closer to out-of-distribution) while UCF101 has a relatively small distribution shift (closer to in-distribution). In Adapt, we fine that there are more context tokens inserted in the image branch compared to text branch on the EuroSAT dataset, whereas the trend is opposite on the UCF dataset.

Adapt removes the constraint that the total context length in the image branch is the same as the total context length in the text branch, and the constraint that each layer has the same context length. The automatically determined binary masks are highly heterogeneous, which is the advantage of Adapt compared to manually designed prompting methods. Besides, context lengths adaptively change on different datasets. It ensures the prompting design is tailored for the specific individual dataset.

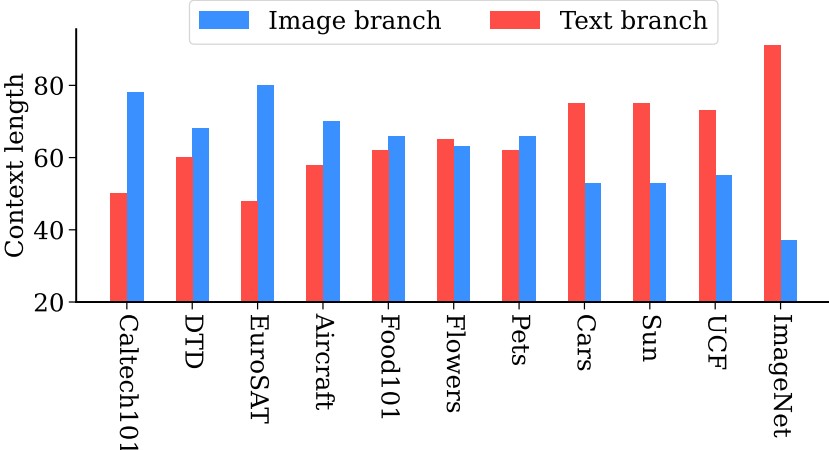

Figure 6: Total context lengths on the text and image branches over 11 datasets. Adapt introduces highly heterogeneous prompts on different datasets and different branches.

## A.9    ADAPTIVE SELECTION OF $\mathcal{T}_{\text{target}}$ IN ADAPT

Ablation study on $\mathcal{T}_{\text{target}}$ shows that the optimal $\mathcal{T}_{\text{target}}$ is different for different datasets as shown in Table 2. Using a universal $\mathcal{T}_{\text{tareget}}$ value as a hyperparameter ensures that the number of trainable parameters is approximately the same across different datasets. On the other hand, adaptive selection of $\mathcal{T}_{\text{target}}$ can lead to different number of trainable parameters on different datasets. To enable customized selection of $\mathcal{T}_{\text{target}}$ for each dataset, we use the validation accuracy as the metric to select the optimal $\mathcal{T}_{\text{target}}$. Specifically, for each dataset, we consider the candidate set $\{\mathcal{T}_{\text{target}}\} = \{32, 64, 128, 256\}$ and we select the model weights with the highest validation accuracy to report the test accuracy.

Table 8 shows the comparison with the Adapt method that allows different $\mathcal{T}_{\text{target}}$ on different datasets. By allowing a dataset-dependent $\mathcal{T}_{\text{target}}$, the performance of the Adapt method boosts.

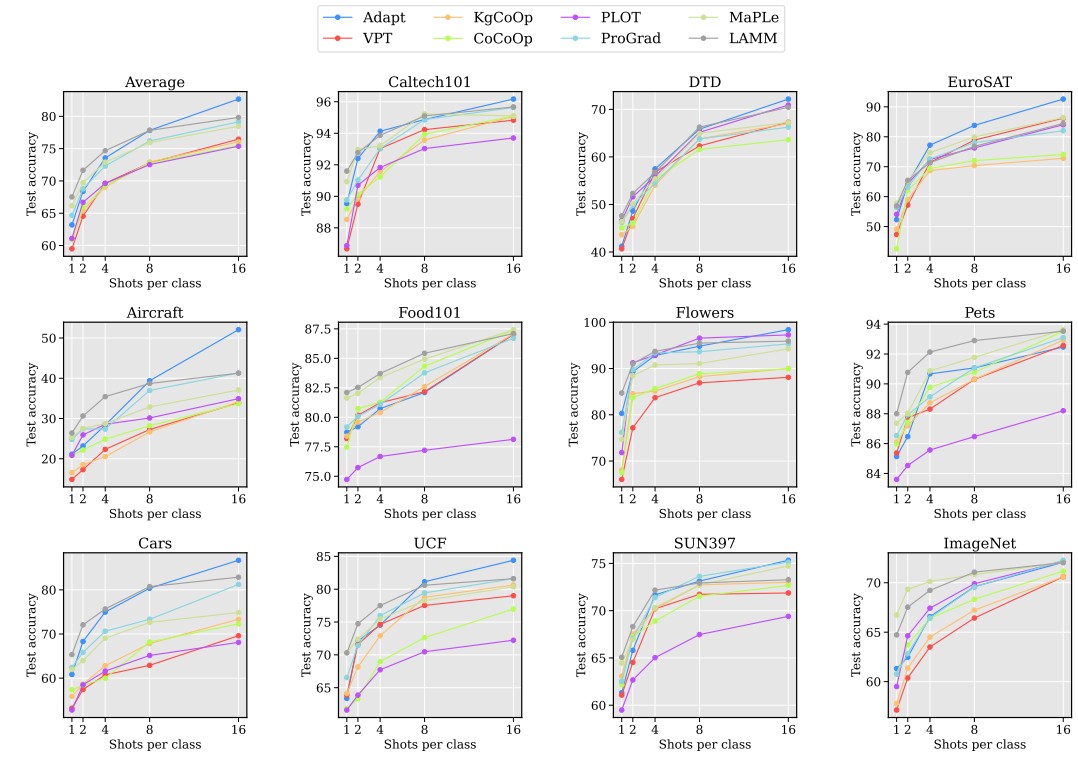

Figure 7: The results of few-shot learning of 1/2/4/8/16 shots on 11 datasets. The average performance is shown on the top left.

Table 8: Performance comparison of the Adapt method with the Adapt method that uses adaptive $\mathcal{T}_{target}$ (Adapt (Adaptive $\mathcal{T}_{target}$)). Adaptive Adapt can have different total context lengths across datasets. Hence, the number of trainable parameters can vary. The validation dataset is used for determining the optimal $\mathcal{T}_{target}$ and the number of epochs at which we obtain the model weights for the test dataset.

| Method | Caltech101 | DTD | EuroSAT | Aircraft | Food101 | Flowers |
|---|---|---|---|---|---|---|
| Adapt ($\mathcal{T}_{target} = 128$) | 95.63 | 72.03 | 92.53 | 50.93 | 83.47 | 97.97 |
| Adapt (Adaptive $\mathcal{T}_{target}$) | 96.17 | 72.17 | 92.60 | 52.07 | 87.03 | 98.40 |

| Method | Pets | Cars | Sun | UCF | ImageNet | Average |
|---|---|---|---|---|---|---|
| Adapt ($\mathcal{T}_{target} = 128$) | 91.07 | 86.17 | 73.67 | 84.40 | 70.83 | 81.70 |
| Adapt (Adaptive $\mathcal{T}_{target}$) | 92.47 | 86.70 | 75.33 | 84.40 | 72.07 | 82.67 |

## A.10 FEW-SHOT LEARNING

We examine the performance of few-shot learning using 1/2/4/8/16 shots. The optimal $\mathcal{T}_{target}$ is determined based on the validation accuracy following Sec. A.9. Figure 7 shows the performance comparison. When the number of shots is small (*e.g.* 1 shot), there is a degradation in the performance. The training process of the Adapt method entails finding the optimal context lengths and optimizing the prompt weights. The highly limited data will impose a challenge in the optimization process. Most existing prompting methods assume that at least 16-shots per cateogry data are available (Hirohashi et al., 2024). When the data is very limited, there is a pronounced performance degradation.

Table 9: Performance comparison of the Adapt methods with and without constraint that the total context lengths on the text and image branches are the same. We use $\mathcal{T}_{\text{target}} = 128$.

| Method | Caltech101 | DTD | EuroSAT | Aircraft | Food101 | Flowers |
|---|---|---|---|---|---|---|
| Adapt w/ constraint | 95.43 | 69.17 | 81.50 | 49.13 | 83.30 | 98.03 |
| Adapt w/o constraint | 95.63 | 72.03 | 92.53 | 50.93 | 83.47 | 97.97 |

| Method | Pets | Cars | Sun | UCF | ImageNet | Average |
|---|---|---|---|---|---|---|
| Adapt w/ constraint | 87.47 | 84.13 | 73.80 | 83.37 | 69.93 | 79.57 |
| Adapt w/o constraint | 91.07 | 86.17 | 73.67 | 84.40 | 70.83 | 81.70 |

Table 10: Performance comparison of 16-shot learning with baselines that apply the coupling function between text and image branches.

| Method | # Trainable Params | GFLOPS | Caltech101 | DTD | EuroSAT | Aircraft | Food101 |
|---|---|---|---|---|---|---|---|
| MaPLe | 3.56 M (2.860%) | 13.35 | 95.10 | 67.27 | 86.40 | 37.07 | **87.43** |
| UPT[†] | 17.78 M (14.305%) | 13.51 | 93.53 | 64.17 | 84.13 | 34.73 | 76.33 |
| Adapt ($\mathcal{T}_{\text{target}} = 128$) | 82,227 (0.066%) | 12.53 | 95.63 | **72.03** | **92.53** | **50.93** | 83.47 |
| Adapt ($\mathcal{T}_{\text{target}} = 32$) | 18,781 (0.015%) | 12.20 | **96.10** | 69.93 | 90.13 | 48.73 | 84.30 |

| Method | Flowers | Pets | Cars | Sun | UCF | ImageNet | Average |
|---|---|---|---|---|---|---|---|
| MaPLe | 94.27 | **93.63** | 74.87 | 74.73 | 80.37 | **72.03** | 78.47 |
| UPT[†] | 95.47 | 90.00 | 74.93 | **75.83** | 78.50 | 70.47 | 76.19 |
| Adapt ($\mathcal{T}_{\text{target}} = 128$) | **97.97** | 91.07 | **86.17** | 73.67 | **84.40** | 70.83 | **81.70** |
| Adapt ($\mathcal{T}_{\text{target}} = 32$) | 97.63 | 90.83 | 85.07 | 74.53 | 84.03 | 71.93 | 81.20 |

† Code is not released for the UPT work. We implement the UPT method on our own. Implementation details are described in Section A.12.

## A.11  CONSTRAINT ON TOTAL CONTEXT LENGTH IN IMAGE AND TEXT BRANCHES

Adapt encourages highly heterogeneous context lengths by allowing different context lengths at different depths and different context lengths in different branches. We examine the performance of the Adapt method with applying the constraint that the total context length in the image branch is the same as that in the text branch. Instead of ranking scores for context tokens in both image and text branches, there are two sets of rankings for the image and text branches, respectively. The selected $\mathcal{T}_{\text{target}}$ is 128.

Table 9 shows the comparison of the Adapt methods with and without constraint. When applying the constraint, there is a remarkable performance degradation, especially for the challenging datasets DTD, EuroSAT and Aircraft. The result supports the motivation of the Adapt method that encourages highly heterogeneous context lengths which are challenging for manually designed prompting methods.

## A.12  COMPARISON WITH BASELINES ALIGNING TEXT AND IMAGE BRANCHES

We compare the Adapt method with prompting methods that have coupling functions between prompts inserted into the text branch and those for the image branch. MaPLe (Khattak et al., 2023) uses a linear transformation function to apply the alignment between text and image branches. UPT (Zang et al., 2022) uses a transformer block to contextualize the prompts for the image branch and text branch.

The code is not released for the UPT method, so we implement the method on our own. Transformer block consisting of the self-attention operator, feed-forward network and layer normalization are used to contextualize the inserted embeddings. The contextualized embedding for each layer of the vision-language model is split for the text and image branches. A fully connected layer per transformer layer of the pre-trained model is used to align the hidden dimension of the embedding to that for the image branch. Please refer UPT (Zang et al., 2022) for more details. The context length is 4 and the prompt depth is 9.

Table 10 shows the performance comparison. Overall, the Adapt method surpasses two baselines by a significant margin. At the same time, introducing the coupling functions inevitably introduces

more trainable parameters and floating-point operations. The number of trainable parameters of MaPLe and UPT methods is significantly higher than the Adapt method.

### A.13 COMPARISON WITH PROMPTING METHOD WITH DIFFERENTIABLE MASKS

Mask tuning (Zheng et al., 2023) applies differentiable masks to model parameters. We use mask tuning for deep prompting methods: we apply differentiable masks to the deep prompts that are consistently inserted and removed (VPT-like prompt), and deep prompts that are inserted only for key and value computation (Adapt-like prompt). The former is the deep prompting method for vision-language models (e.g. VPT (Jia et al., 2022) and MaPLe (Khattak et al., 2023)).

Table 11 shows the performance comparison. Masking tuning does not apply the constraint on the number of trainable parameters while the Adapt method uses $\mathcal{T}_{\text{target}}$ to ensure the number of trainable parameters is approximately the same on different datasets. For a fair comparison, we include the result that adaptively changes $\mathcal{T}_{\text{target}}$ on different datasets denoted as Adapt (Adaptive $\mathcal{T}_{\text{target}}$). The performance of the Adapt method is better than directly applying differentiable masks to the prompt tuning.

Table 11: Performance comparison of applying mask tuning (Zheng et al., 2023) to prompting methods. We apply mask tuning for two different prompting methods: VPT-like deep prompting and Adapt-like deep prompting.

| Method | Caltech101 | DTD | EuroSAT | Aircraft | Food101 | Flowers |
|---|---|---|---|---|---|---|
| Mask tuning + VPT-like deep prompting | 93.60 | 64.13 | 71.80 | 33.57 | 84.30 | 88.73 |
| Mask tuning + Adapt-like deep prompting | 95.53 | 71.87 | 92.17 | 51.03 | 82.17 | 97.43 |
| Adapt ($\mathcal{T}_{\text{target}} = 128$) | 95.63 | 72.03 | 92.53 | 50.93 | 83.47 | 97.97 |
| Adapt (Adaptive $\mathcal{T}_{\text{target}}$) | 96.17 | 72.17 | 92.60 | 52.07 | 87.03 | 98.40 |

| Method | Pets | Cars | Sun | UCF | ImageNet | Average |
|---|---|---|---|---|---|---|
| Mask tuning + VPT-like Deep prompting | 90.37 | 71.73 | 71.17 | 73.63 | 70.37 | 73.95 |
| Mask tuning + Adapt-like deep prompting | 88.90 | 85.70 | 73.03 | 83.73 | 70.03 | 81.05 |
| Adapt ($\mathcal{T}_{\text{target}} = 128$) | 91.07 | 86.17 | 73.67 | 84.40 | 70.83 | 81.70 |
| Adapt (Adaptive $\mathcal{T}_{\text{target}}$) | 92.47 | 86.70 | 75.33 | 84.40 | 72.07 | 82.67 |

### A.14 APPLYING ADAPT TO LANGUAGE MODELS

In addition to vision-language models, we apply the Adapt method to language models. The training process is the same as Algorithm 1 except that pruning happens in the monomodality. A fixed $\mathcal{T}_{\text{target}} = 128$ is used to ensure the prompt complexity is the same on different datasets. The pre-trained model is BERT (Kenton & Toutanova, 2019). The baseline we choose is P-Tuning v2 (Liu et al., 2021). We use the default hyperparameters for the P-Tuning v2 method. Datasets are COPA, BoolQ and RTE from SUPERGlue benchmark (Wang et al., 2019).

Table 12 shows the comparison of the Adapt method with the baseline method on 3 different datasets. We observe that the Adapt method can achieve higher test accuracy with a smaller number of trainable parameters.

Table 12: Performance comparison of applying Adapt to the BERT model. $\mathcal{T}_{\text{target}} = 128$ is used in the Adapt method.

| Method | COPA | | BoolQ | | RTE | |
|---|---|---|---|---|---|---|
| | # parameters | Accuracy | # parameters | Accuracy | # parameters | Accuracy |
| P-Tuning v2 | 0.787 M | 78.00 | 1.968 M | 75.02 | 0.985 M | 78.17 |
| Adapt | 0.297 M | 80.00 | 0.297 M | 76.50 | 0.297 M | 79.17 |

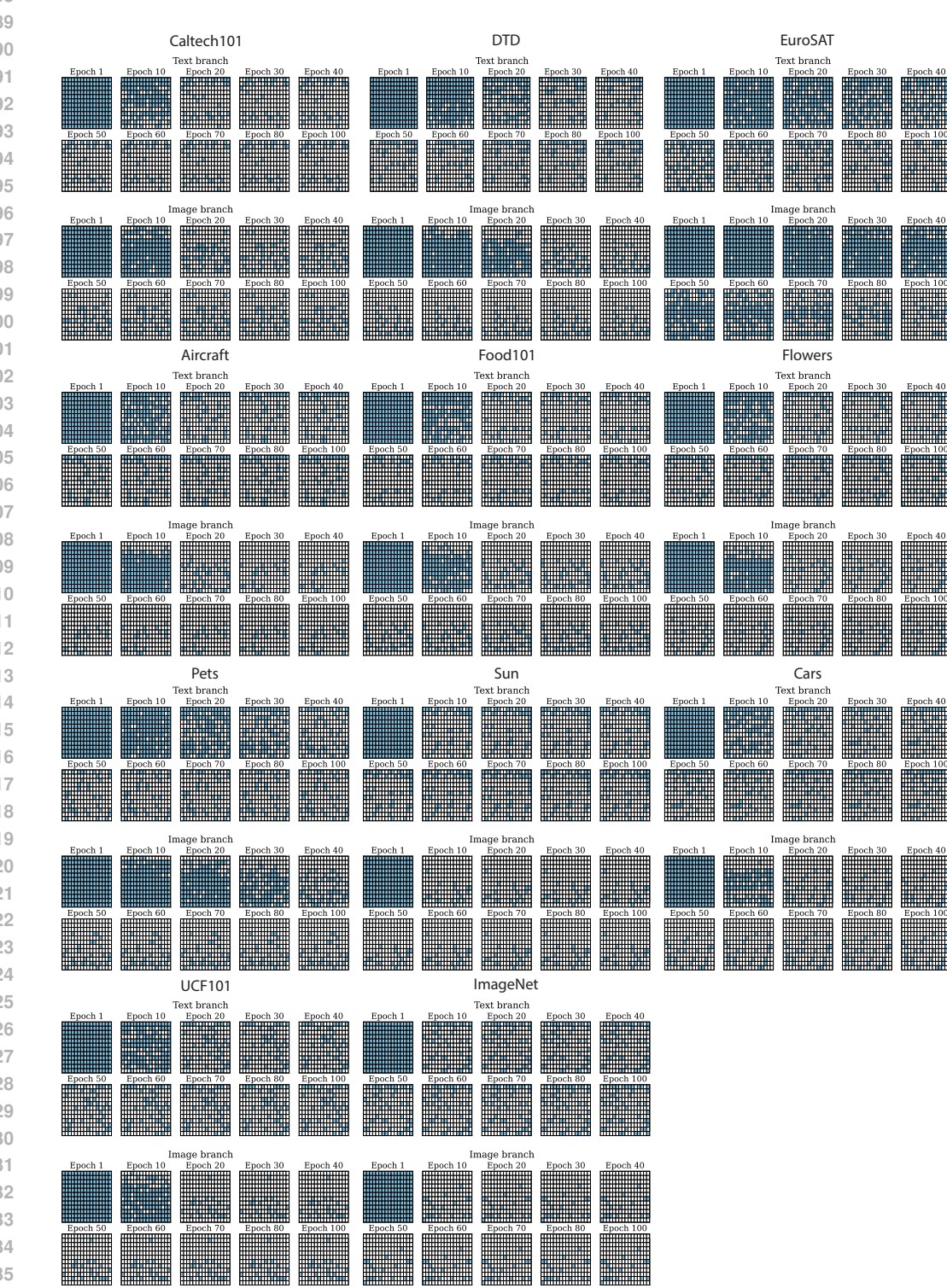

Figure 8: Pruning process of binary masks for the text and image branches for 11 datasets. $\mathcal{T}_{\text{target}} = 64$ and the warmup epoch is 5. Owing to the warmup process, $\mathcal{M}(t)$ at the epoch of 1 is same as the $\mathcal{M}(0)$. We set $\mathcal{T}_{\text{target}}$ so that the pruned $\mathcal{M}(t)$ after convergence is a sparse matrix.

