# OpenReview forum: "ADAPT: Adaptive Prompt Tuning for Pre-Trained Vision-Language Models"
_ICLR.cc/2025/Conference — Submitted to ICLR 2025_

### Official Review · Reviewer_Xqnt · 2024-10-24

**Soundness:** 3
**Presentation:** 1
**Contribution:** 3
**Rating:** 6
**Confidence:** 4

**Summary:**

In this paper, the authors propose adaptively pruning prompt tokens during the prompt tuning, rather than using fixed prompt length. They use metrics in network pruning to compute the importance scores of prompt tokens and prune less important tokens gradually.

**Strengths:**

Strength:
+The average performance on 11 downstream datasets verifies the effectiveness of the proposed methods.
+The proposed method shows slightly fewer FLOPs than existing methods.
+Adaptively changing the prompt tokens is an interesting idea.

**Weaknesses:**

Weakness:
1. There is more than one page to write. It looks like a paper in progress  The authors should consider to include more experiments and analysis. For example, the authors can show that different datasets prefer different prompt token lengths to verify the importance of the proposed method.

2. In line 377, the authors write “The result is shown in Appendix Figure 4. However, the appendix is missing. The authors should move it from the supplementary material to the end of the main paper.

3. How do we determine the number of tokens to prune each each layer?

4. How to set the number of prune steps rp.

5. There are too many mathematical symbols, especially in Algorithm 1, making it hard to understand, even though the operation used in this paper is easy. The authors should improve this to improve the readability.

6. There are only two paragraphs in the Introduction Section. The authors should consider splitting them into more paragraphs.

7. The proposed methods are highly related to dynamic neural networks. The authors should discuss it and cite related papers.

I think that the idea of this paper is good enough. However, the authors should improve their presentation.


Issues:
In Figure1, the authors should indicate the proposed method with “Adapt (Ours)”.

**Questions:**

See Weakness

---

> ### Author Response · Authors · 2024-11-21
>
> The authors thank the reviewer for the feedback. Here is one-to-one response:
>
> - Weakness 1: The ICLR submission rule suggests the total number of pages to be 9. Following the reviewer’s suggestion, we add the analysis of the total context lengths in Appendix A.8.
>
> We believe the comment “There is more than one page to write.” is a misunderstanding. In the page length section of ICLR call for papers, it is stated that “We encourage authors to be crisp in their writing by submitting papers with 9 pages of main text. We recommend that authors only use the longer page limit in order to include larger and more detailed figures. However, authors are free to use the pages as they wish, as long as they obey the page limits.” (please refer to the submission guidance https://iclr.cc/Conferences/2025/CallForPapers).
>
> In our original submission, we show the result of pruned binary masks in Appendix Fig. 8. The results indicate that the context lengths are highly heterogeneous. We added a section (Appendix A.8) to analyze the total context lengths in text and image branches. The total context lengths are highly related to datasets as shown in Appendix Figure 6.
>
> - Weakness 2: We believe this is an oversight. We did upload the appendix with our original submission. Please refer to the link for supplementary material.
>
> - Weakness 3: The number of tokens to be pruned depends on the importance of the tokens. We use a scoring function to evaluate the importance of tokens. We do not constrain the number of tokens per layer. Please refer to Algorithm 1 for more details. If the context length for a certain layer is 0, it means there is no prompt for this layer.
>
> - Weakness 4: Prune rate $r_p$ is a hyperparameter. We show the result of using different pruning rates in Appendix A.3. The result suggests that different pruning rates do not significantly change the final mask, which indicates the robustness of the pruning strategy.
>
> - Weakness 5: We added more text explanations for the algorithm at the bottom of page 5 and text descriptions in the Algorithm 1 (line 7, 8, 9 and 12).
>
> - Weakness 6: We split the paragraphs in Introduction into multiple paragraphs. The first paragraph gives an overview of the PEFT methods and introduces prompting methods. The second paragraph elaborates on the two categories of prompting methods. The third paragraph talks about works related to continuous prompts. The fourth paragraph talks about the motivation why we propose our method. The fifth paragraph gives the whole picture of the Adapt method.
>
> - Weakness 7: We are not sure which dynamic neural networks the reviewer is referring to. Can the reviewer give some related references?
>
> We checked classic dynamic neural network references. One line of work focuses on dynamic neural architecture. The depth [1-2], width [3-4] and structure [5] can change. Adapt uses the pre-trained model and the model weights and structure are fixed during the training process. Another line of work focuses on dynamic parameters. The model parameters such as kernel shape [6], channels [7] and features [8] can change. Adapt changes the context lengths during the training process, which do not affect the model parameters or structures. Dynamic neural networks generally use differentiable parameters to adaptively change parameters. The parameters in Adapt are not determined in a differentiable way. It changes in a discrete way.
>
> We thank the reviewer’s suggestion. Here is what we did to improve the presentation:
>
> 1. We changed “Adapt” to “Adapt (ours)” in Figure 1
> 2. We split the paragraphs in the Introduction into multiple paragraphs
> 3. We added one paragraph to explain more details in Algorithm 1 and text descriptions within Algorithm 1
> 4. We added the analysis of total context lengths in Appendix A.8
>
> - Issue: We thank the reviewer’s suggestion. We changed “Adapt” to “Adapt (ours)” in Figure 1.

---

> > ### Author Response · Authors · 2024-11-21
> >
> > [1] Wang, Xin, et al. "Skipnet: Learning dynamic routing in convolutional networks." Proceedings of the European conference on computer vision (ECCV). 2018.
> >
> > [2] Veit, Andreas, and Serge Belongie. "Convolutional networks with adaptive inference graphs." Proceedings of the European conference on computer vision (ECCV). 2018.
> >
> > [3] Bengio, Yoshua, Nicholas Léonard, and Aaron Courville. "Estimating or propagating gradients through stochastic neurons for conditional computation." arXiv preprint arXiv:1308.3432 (2013).
> >
> > [4] Mullapudi, Ravi Teja, et al. "Hydranets: Specialized dynamic architectures for efficient inference." Proceedings of the IEEE Conference on Computer Vision and Pattern Recognition. 2018.
> >
> > [5] Huang, Gao, et al. "Multi-scale dense networks for resource efficient image classification." arXiv preprint arXiv:1703.09844 (2017).
> >
> > [6] Gao, Hang, et al. "Deformable kernels: Adapting effective receptive fields for object deformation." arXiv preprint arXiv:1910.02940 (2019).
> >
> > [7] Yang, Brandon, et al. "Condconv: Conditionally parameterized convolutions for efficient inference." Advances in neural information processing systems 32 (2019).
> >
> > [8] Su, Hang, et al. "Pixel-adaptive convolutional neural networks." Proceedings of the IEEE/CVF Conference on Computer Vision and Pattern Recognition. 2019.

---

> > ### Comment · Reviewer_Xqnt · 2024-11-24
> >
> > Thanks the responses from the authors.
> >
> > Weakness 2: I recommend including this content directly within the PDF after the main paper, rather than placing it in the "Supplementary Material" section. This change could improve the overall readability and accessibility of the paper.
> >
> > Weakness 7: Thank you for the detailed review of dynamic neural networks. I now understand that the proposed method differs from traditional neural networks. However, I am curious whether the proposed method has any relevance to dynamic sparse training [1], which is also designed to address the over-parameterization problem. Additionally, I still believe the literature review in this paper could be further improved.
> >
> > [1] Chasing Sparsity in Vision Transformers: An End-to-End Exploration.

---

> > > ### Author Response · Authors · 2024-11-25
> > >
> > > The authors thank the reviewer for the feedback. Here is the one-to-one response:
> > >
> > > Weakness 2: Following the suggestion, we combined main paper and appendix in the same PDF file. The new file has been uploaded.
> > >
> > > Weakness 7: We thank the reviewer for the clarification and the discussion on dynamic spars training. We added a paragraph named “Sparse Training” in Section 2 Related Work. Please refer to the updated manuscript.
> > >
> > > In the level of dynamic training, our method is similar to dynamic sparse training. But there are some key differences:
> > >
> > > 1. Our method works on soft prompts while dynamic sparse training works on model parameters, specifically, the weights of linear transformation layers for query, key and value.
> > >
> > > 2. Dynamic sparse training uses element-wise multiplication for model weights while we use masks to select important soft prompts.  The selection will cause different context lengths at different depths while element-wise multiplication does not change context lengths at different depths.
> > >
> > > 3. Dynamic sparse training considers sparsity distribution to avoid “over-pruning” that might cause disconnectivity of neural layers. In our method, if all context tokens for one transformer layer are pruned, it indicates that prompt depth decreases by one. There is no “over-pruning” issue in our method.

---

> > > > ### Comment · Reviewer_Xqnt · 2024-11-26
> > > >
> > > > Thanks the responses from the authors. I raise the score to 6.

---

> ### Author Response · Authors · 2024-11-26
>
> We thank the reviewer for the thoughtful feedback and for recognizing the strengths of our work. We are especially grateful for the positive assessment and the raised scores

---

### Official Review · Reviewer_bqgw · 2024-11-03

**Soundness:** 2
**Presentation:** 3
**Contribution:** 2
**Rating:** 5
**Confidence:** 2

**Summary:**

The paper assumes that a fixed context length for prompts may lead to either redundant context tokens or insufficient context length when transferring a pre-trained model to downstream tasks. Based on this assumption, the paper proposes a method to automatically determine the prompt length.

**Strengths:**

The proposed method, ADAPT, changes the context lengths for different transformer layers by iteratively pruning context tokens. ADAPT surpasses the SOTA method on 16-shot image classification tasks.

**Weaknesses:**

It is unclear why the convergence of model training is determined solely by reaching T_target. T_target may vary across different training datasets, but it is set to a fixed value for all datasets. Additionally, if the mask for the text encoder is too sparse, this training target might restrict the sparsity of the mask for the image encoder.

The paper should provide a more detailed analysis of the learned binary masks. According to Figure 3, on the EuroSAT dataset, more context tokens are required in the middle layer of the image encoder, while the first layer of the text encoder requires more context tokens. An analysis of this discrepancy should be included.

ADAPT is trained and evaluated on the few-shot classification task, following the CoOP methodology. Thus, it should also report results under other training settings (1-shot, 2-shot, 4-shot, and 8-shot) to enable a more comprehensive comparison with state-of-the-art methods.

Moreover, UPT[1] should be included for comparison, as it also introduces prompts in both the text and image encoders, similar to ADAPT.

[1] Unified vision and language prompt learning.

**Questions:**

Please see the questions in Weaknesses.

---

> ### Author Response · Authors · 2024-11-21
>
> The authors thank the reviewer for the feedback. Here is one-to-one response:
>
> - Weakness 1: Originally, we set a fixed $T_{target}$ to ensure that on different datasets, Adapt has a similar model complexity and is fair to the baselines. The reviewer is correct that by making $T_{target}$ adaptive to each dataset, the performance can be further improved. To demonstrate this point, we consider the setting allowing the dataset-dependent $T_{target}$. The details are reported in Appendix A.9, the performance comparison is:
>
> | Method | Caltech101 | DTD | EuroSAT | Aircraft | Food101 | Flowers | Pets | Cars | Sun | UCF | ImageNet | Average |
> | --- | --- | --- | --- | --- | --- | --- | --- | --- | --- | --- | --- | --- |
> | Adapt | 95.63 | 72.03 | 92.53 | 50.93 | 83.47 | 97.97 | 91.07 | 86.17 | 73.67 | 84.40 | 70.83 | 81.70 |
> | Adapt (adaptive $T_{target}$) | 96.17 | 72.17 | 92.60 | 52.07 | 87.03 | 98.40 | 92.47 | 86.70 | 75.33 | 84.40 | 72.07 | 82.67 |
>
> - Weakness 2: What the reviewer points out is the uniqueness of the Adapt method that introduces the heterogeneous context lengths. We analyzed the context lengths on different datasets. The results are added in Appendix A.8. We noticed that for datasets that have a larger degree of out-of-distribution, there will be more context tokens added to the image branch.
>
> - Weakness 3: We added results using 1/2/4/8-shot. We use the validation dataset to determine the optimal $T_{target}$ named Adapt (Adaptive $T_{target}$). Details regarding  Adapt (Adaptive $T_{target}$) are shown in Appendix A.9 and the result of 1/2/4/8/16-shot is shown in Appendix A.10.
>
> - Weakness 4: We wanted to include UPT work for the comparison, but the code is not included in the repo [1]. We implemented UPT [2] on our own. Below is the performance we got:
>
> | Method  | Caltech101 | DTD | EuroSAT | Aircraft | Food101 | Flowers | Pets | Cars | Sun | UCF | ImageNet | Average |
> | --- | --- | --- | --- | --- | --- | --- | --- | --- | --- | --- | --- | --- |
> | UPT  | 93.53 | 64.17 | 84.13 | 34.73 | 76.33 | 95.47 | 90.00 | 74.93 | 75.83 | 78.50 | 70.47 | 76.19 |
> | Adapt | 95.63 | 72.03 | 92.53 | 50.93 | 83.47 | 97.97 | 91.07 | 86.17 | 73.67 | 84.40 | 70.83 | 81.70 |
>
> We have added the comparison in Appendix A.12 and cited UPT work in the updated version.
>
> [1] https://github.com/yuhangzang/UPT
>
> [2] Zang, Yuhang, et al. "Unified vision and language prompt learning." arXiv preprint arXiv:2210.07225 (2022).

---

> ### Comment · Area_Chair_yWcZ · 2024-11-25
>
> Dear Reviewer bqgw,
>
> Could you kindly review the rebuttal thoroughly and let us know whether the authors have adequately addressed the issues raised or if you have any further questions.
>
> Best,
>
> AC of Submission8374

---

> ### Comment · Reviewer_bqgw · 2024-11-26
>
> Thanks for the responses. The rebuttal shows that $T_{target}$ should be set to different values on different datasets. This suggests that automating the selection of $T_{target}$ would be a valuable improvement.  Additionally, the performance of ADAPT in other training setting is lower than that of the state-of-the-art methods. Therefore, I maintain my initial score.

---

> > ### Author Response · Authors · 2024-11-27
> >
> > Thank you for the response. We are glad that most of your concerns were addressed. Regarding the adaptive $T_{target}$ comment, we want to emphasize that our method already has the best average performance than all baselines even using fixed $T_{target}$, as shown in Table 1 in the manuscript. In the rebuttal, we show that by allowing the $T_{target}$ value to be adaptive for each dataset, the performance can be further improved because of the additional design choices in our search algorithm. This new result should not be misinterpreted as the ineffectiveness of the fixed-value result, given that the search spaces are different. One search space has the constraint that the total number of context lengths is fixed over all datasets while the other one does not have such constraint

---

> ### Author Response · Authors · 2024-12-01
>
> We thank the reviewer for the time and effort spent in evaluating our work. As the author-reviewer discussion phase is ending soon, we would like to kindly ask if the concern has been addressed with our latest response (https://openreview.net/forum?id=1L9vdc7BB5&noteId=PVCF08vTPX)?

---

### Official Review · Reviewer_qkws · 2024-11-03

**Soundness:** 2
**Presentation:** 3
**Contribution:** 2
**Rating:** 5
**Confidence:** 5

**Summary:**

This paper proposes a deep continuous prompting method dubbed Adapt that encourages heterogeneous context lengths.

**Strengths:**

-The paper is well-written.

-Extensive experiments on 11 downstream datasets reveal the advantage of Adapt.

-Adding mask to the prompts of different depth is an interesting idea.

**Weaknesses:**

Adding learnable mask to the prompts of different depth is an interesting idea. But, existing methods [1] proposed to add learnable mask to the parameters of CLIP. Adding learnable mask to parameters and add learnable mask to prompt have similar methods. Moreover, this paper did not discuss the difference between ADAPT and [1], which miss this key reference.

[1] Regularized Mask Tuning: Uncovering Hidden Knowledge in Pre-trained Vision-Language Models, ICCV 2023

**Questions:**

-The hyperparameter T_target controls sparsity of masks. According to Table 2, the model reaches better averaged performance when T_target is set to a larger value (the masks are less sparse). What if T_target is set to a value larger than 128? What is the upper bound of the proposed method?

-Ablations on prompt depth and context length should be conducted.

-To demonstrate the effectiveness of the proposed method on few-shot classification tasks, the paper should provide results on 1/2/4/8-shot training setting, similar to those reported in CoOP and other related studies.

---

> ### Author Response · Authors · 2024-11-21
>
> The authors thank the reviewer for the comments. Here is one-to-one response:
>
> - Weakness 1: We would like to start by clarifying that reference [1] only focuses on applying differentiable masks to model weights, which is different from our soft prompt learning setup. The differences between reference [1] and Adapt are:
>
> 1. [1] uses differentiable masks while Adapt does not use differentiable masks. The variation of masks in [1] depends on the loss function and threshold function. Our method changes masks by pruning process and scoring functions. Hence, using differentiable masks introduces additional trainable parameters.
> 2. [1] applies masks to the model parameters (e.g. MSHA layer parameters) while Adapt applies masks to soft prompts.
> 3. [1] uses the mask to **zero out** some model parameters (element-wise multiplication) while Adapt uses the mask to **select** valid context tokens and there will be no zero-out tokens. When applying [1] to deep prompts, the context length **does not change**. When applying Adapt, the context length **does change**.
>
> Following the reviewer's suggestion, we did experiments using differentiable masks with the closest possible setup. Since deep prompting methods for vision-language models such as MaPLe [2] and VPT [3] do not insert the deep prompts in the same way as we do (comparison is shown in Figure 2 (a) and (c)), we tested the differentiable mask + traditional deep prompting denoted as “VPT + Differentiable Mask” (prompts are inserted for query, key and value and removed after self-attention) and the differentiable mask + our deep prompting method denoted as “Adapt + Differentiable Mask” (prompts are inserted for only key and value computation, no inserted prompts are removed after self-attention).
>
> Differentiable masks do not apply a constraint on the total context length, so the prompt complexity can be different on different datasets. For a fair comparison, we include the performance of Adapt (adaptive $T_{target}$) that uses optimal $T_{target}$ determined by the validation dataset (details are described in Appendix A.9). In differentiable mask methods, the threshold alpha for the binary function I(M > alpha) is set to be 0.5. We do find a performance degradation using differentiable masks, especially using the traditional deep prompting method (a comparison between traditional deep prompting and our deep prompting methods is shown in Figure 2 (a) and (c)). The performance comparison is shown below:
>
> | Method | Caltech101 | DTD | EuroSAT | Aircraft | Food101 | Flowers | Pets | Cars | Sun | UCF | ImageNet | Average |
> | --- | --- | --- | --- | --- | --- | --- | --- | --- | --- | --- | --- | --- |
> | Adapt | 95.63 | 72.03 | 92.53 | 50.93 | 83.47 | 97.97 | 91.07 | 86.17 | 73.67 | 84.40 | 70.83 | 81.70 |
> | Adapt (Adaptive $T_{target}$) | 96.17 | 72.17 | 92.60 | 52.07 | 87.03 | 98.40 | 92.47 | 86.70 | 75.33 | 84.40 | 72.07 | 82.67 |
> | Adapt  + Differentiable Mask | 95.53 | 71.87 | 92.17 | 51.03 | 82.17 | 97.43 | 88.90 | 85.70 | 73.03 | 83.73 | 70.03 | 81.05 |
> | VPT + Differentiable Mask    | 93.60 | 64.13 | 71.80 | 33.57 | 84.30 | 88.73 | 90.37 | 71.73 | 71.17 | 73.63 | 70.37 | 73.95 |
>
> We have added this new result and cited [1] in Appendix A.13.
>
> - Question 1: We examine the performance of using $T_{target} = 256$ and $T_{target} = 512$. The new results are included and marked in blue color in Table 2 of the updated manuscript. The results empirically indicate that the upper bound for the average accuracy is 81.70%.
>
> - Question 2: In Adapt, prompt depth and context length are automatically determined. Hence, they are not hyperparameters of our model. The hyperparameters used by Adapt is $T_{target}$, $r_p$ (pruning rate), and accumulation steps (number of steps to compute the accumulated scores). The effect of $T_{target}$ is reported in Table 2. The effect of $r_p$ is reported in Appendix A.3.
>
> Please refer to line 13 in Algorithm 1 and Figure 8 in Appendix. The prompt depth and context length are automatically determined.
>
> - Question 3: We add results using 1/2/4/8-shot training setting in Appendix A.10 using Adaptive $T_{target}$. The Adaptive $T_{target}$ is reported in Appendix A.9 and The few-shot learning result is shown in Appendix A.10.
>
> [1] Zheng, Kecheng, et al. "Regularized mask tuning: Uncovering hidden knowledge in pre-trained vision-language models." Proceedings of the IEEE/CVF International Conference on Computer Vision. 2023.
>
> [2] Khattak, Muhammad Uzair, et al. "Maple: Multi-modal prompt learning." Proceedings of the IEEE/CVF Conference on Computer Vision and Pattern Recognition. 2023.
>
> [3] Jia, Menglin, et al. "Visual prompt tuning." European Conference on Computer Vision. Cham: Springer Nature Switzerland, 2022.

---

> ### Comment · Area_Chair_yWcZ · 2024-11-25
>
> Dear Reviewer qkws,
>
> Could you kindly review the rebuttal thoroughly and let us know whether the authors have adequately addressed the issues raised or if you have any further questions.
>
> Best,
>
> AC of Submission8374

---

> ### Comment · Reviewer_qkws · 2024-11-26
> **Official Comment by Reviewer qkws**
>
> I thank the authors for their response.
>
> Some concerns have been addressed. I appreciate this work, but I still believe that the core idea is similar to [1], these two methods both add Binary Mask to some parameters (e.g., model weights or prompt weights).
>
> Additionally, the ADAPT does not show the advantages in few-shot learning tasks (1/2/4/8). As the number of available images decreases (from 16 shot to 1 shot), the performance advantage of this method becomes inferior to existing methods (e.g., MaPLe\MaPLe\LAMM). This indicates that the method is highly dependent on the amount of data (more than 8 shots) and is not very robust.
>
> Regarding method innovation and robustness, I think there is still some room for improvement in this work. I will keep my original score of 3.

---

> > ### Author Response · Authors · 2024-11-26
> >
> > We respectfully disagree that our method has low novelty due to the similar notion of "binary mask" in [1]". We argue that making this claim is like saying many works in binary mask pruning/learning have no novelty because of the similarity to dropout, for which we hope the reviewer would disagree. The differences compared to baseline methods are listed in the reply (https://openreview.net/forum?id=1L9vdc7BB5&noteId=z00GyztQeQ).
> >
> > As evidence, we find highly impact papers using binary masks:
> >
> > - [2] (ICLR 2019) uses binary masks and proposes Lottery Ticket Hypothesis (LTH).
> > - [3] (ICLR 2023) uses binary masks through the lens of Ramanujan Graph.
> > - [4] (ICML 2022) uses binary masks based on LTH.
> > - [5] (JMLR 2021) uses binary masks in the sparse training.
> > - [6] (ICLR 2023) uses binary masks in large language models.
> > - [7] (NIPS 2020) uses binary masks. The pruning process is based on the proposed criteria.
> > - [8] (ICLR 2024) uses binary masks to accelerate pre-training process of large language models.
> >
> > Reviews/Surveys on network pruning and spare training [A1-A4] list a variety of papers using binary masks. We notice that none of them explore the application of network pruning in prompts. Directly applying element-wise multiplication will cause issues in prompting method as it generates zero embeddings in prompts. Please refer to our experiments in Appendix Table 11.
> >
> > [1] Zheng, Kecheng, et al. "Regularized mask tuning: Uncovering hidden knowledge in pre-trained vision-language models." Proceedings of the IEEE/CVF International Conference on Computer Vision. 2023.
> >
> > [2] Frankle, Jonathan, and Michael Carbin. "The lottery ticket hypothesis: Finding sparse, trainable neural networks." ICLR 2019 (2019).
> >
> > [3] Hoang, Duc NM, and Shiwei Liu. "Revisiting pruning at initialization through the lens of ramanujan graph." ICLR 2023 (2023).
> >
> > [4] Pal, Bithika, et al. "A study on the ramanujan graph property of winning lottery tickets." International Conference on Machine Learning. PMLR, 2022.
> >
> > [5] Hoefler, Torsten, et al. "Sparsity in deep learning: Pruning and growth for efficient inference and training in neural networks." Journal of Machine Learning Research 22.241 (2021): 1-124.
> >
> > [6] Sun, Mingjie, et al. "A simple and effective pruning approach for large language models." arXiv preprint arXiv:2306.11695 (2023).
> > [7] Tanaka, Hidenori, et al. "Pruning neural networks without any data by iteratively conserving synaptic flow." Advances in neural information processing systems 33 (2020): 6377-6389.
> >
> > [8] Xia, Mengzhou, et al. "Sheared llama: Accelerating language model pre-training via structured pruning." arXiv preprint arXiv:2310.06694 (2023).
> >
> > [A1] Cheng, Hongrong, Miao Zhang, and Javen Qinfeng Shi. "A survey on deep neural network pruning: Taxonomy, comparison, analysis, and recommendations." IEEE Transactions on Pattern Analysis and Machine Intelligence (2024).
> >
> > [A2] He, Yang, and Lingao Xiao. "Structured pruning for deep convolutional neural networks: A survey." IEEE transactions on pattern analysis and machine intelligence (2023).
> >
> > [A3] Fedus, William, Jeff Dean, and Barret Zoph. "A review of sparse expert models in deep learning." arXiv preprint arXiv:2209.01667 (2022).
> >
> > [A4] Qiao, Lin-bo, et al. "A systematic review of structured sparse learning." Frontiers of Information Technology & Electronic Engineering 18.4 (2017): 445-463.

---

> ### Comment · Reviewer_qkws · 2024-11-27
> **Official Comment by Reviewer qkws**
>
> Thanks for your response.
>
> I greatly appreciate the author's design in the subfield (i.e. using binary masks to solve how CLIP can be used for downstream task training, such as few shot) that utilizes binary masks effectively. We would like to clarify my viewpoint again. I am not claiming that ADAPT has low novelty due to the similar notion of "binary mask". But rather, in this subfield, the use of binary masks for trasferring CLIP to few-shot tasks has already been explored, and I think the core idea 'use binary mask for transferring CLIP to few-shot' is similar. The authors claim that 'this is the first work to prune prompts'. If the core idea is to 'use binary mask in prompts', prompt learning has a wide rang e of applications and should be validated for its effectiveness in various tasks, such as prompt learning in LLM.
>
> Additionally, the ADAPT is highly dependent on the amount of data (more than 8 shots). As shown in Figure 1, the authors only show the results in 16 shot, but in 1/2/4/8 shot setting the performance has significantly decreased. In few-shot setting of CLIP, 1/2/4/8 shot settings are also important.
>
> The initial version did not provide a detailed comparison with the binary mask scheme used in the CLIP field in related work and experiments. I find that the updated version provided these and can be improved by more detailed comparision and related works. After carefully reading the updated version, I will raise my score to 5 based on the Sparse Training of related work and comparision with differential mask from the authors. But I am still concerned about performance degradation with smaller shot.

---

> > ### Author Response · Authors · 2024-11-27
> >
> > We thank the reviewer for the response and appreciate that the reviewer decided to raise the score. Your comments and suggestions are tremendous in helping us to better reflect our contributions and findings.
> >
> > Following your comment on the limitation of the few-shot setting, we added a new paragraph about the limitation of the Adapt method in Section 5 of the updated version.
> >
> > We also agree with your comment that showing Adapt is beneficial to language models can further demonstrate its impact and effectiveness. Due to the time limit, we did preliminary experiments on applying Adapt for BERT [1]. The baseline is p-tuning v2 [2]. Consistent with the CLIP results, we observed a performance gain compared to p-tuning v2. The performance comparison is shown in the Table below. Details are reported in Appendix A.14.
> >
> > | Method | COPA # param | COPA Acc | BoolQ # param | BoolQ Acc | RTE # param | RTE Acc |
> > | --- | --- | --- | --- | --- | --- | --- |
> > | P-Tuning v2 | 0.787 M | 78.00 | 1.968 M | 75.02 | 0.985 M | 78.17 |
> > | Adapt | 0.297 M | 80.00 | 0.297 M | 76.50 | 0.297 M | 79.17 |
> >
> > [1] Kenton, Jacob Devlin Ming-Wei Chang, and Lee Kristina Toutanova. "Bert: Pre-training of deep bidirectional transformers for language understanding." Proceedings of naacL-HLT. Vol. 1. 2019.
> >
> > [2] Liu, Xiao, et al. "P-tuning v2: Prompt tuning can be comparable to fine-tuning universally across scales and tasks." arXiv preprint arXiv:2110.07602 (2021).

---

> > > ### Author Response · Authors · 2024-12-01
> > >
> > > We thank the reviewer for the time and effort spent in evaluating our work. As the author-reviewer discussion phase is ending soon, we would like to kindly ask if there is any feedback based on our latest response (https://openreview.net/forum?id=1L9vdc7BB5&noteId=c8QFuekBLV).

---

> > > > ### Comment · Reviewer_qkws · 2024-12-03
> > > > **Official Comment by Reviewer qkws**
> > > >
> > > > Thank you for your response and the experiment of prompt learning in LLM.
> > > >
> > > > After reading the rebuttal and other reviewers' comments, I will raise my score to 5. But this paper still has room for improvement (*e.g.*, the method effectiveness in 1/2/4/8 shot settings, and more related works in 'Sparse Training' section).

---

### Official Review · Reviewer_Tr3E · 2024-11-08

**Soundness:** 2
**Presentation:** 2
**Contribution:** 2
**Rating:** 6
**Confidence:** 5

**Summary:**

To address the limitations of fixed-length prompt tuning approaches for pre-trained vision-language models, the authors propose ADAPT, an adaptive prompt tuning method that dynamically determines optimal prompt lengths during fine-tuning. By employing an iterative pruning strategy, ADAPT identifies and removes less relevant prompt tokens at each layer, allowing efficient parameter usage while maintaining model performance. The authors evaluate ADAPT across 11 benchmark datasets, demonstrating that the method significantly reduces the number of parameters required while achieving competitive or improved accuracy. This adaptive approach highlights the benefits of automatic context length adjustment compared to manually designed fixed-length prompts.

**Strengths:**

The authors propose a novel adaptive prompt tuning approach, ADAPT, that effectively reduces the number of parameters needed for pre-trained vision-language models while maintaining competitive performance across a variety of downstream tasks. This efficiency is a notable contribution to prompt-based fine-tuning methods.
By leveraging an iterative pruning mechanism, ADAPT dynamically adjusts the prompt lengths for different layers, enabling a flexible solution that outperforms traditional fixed-length prompt tuning methods, particularly in scenarios that require task-specific adaptations.
The approach is validated on 11 diverse datasets, covering different vision-language tasks. This broad evaluation demonstrates the adaptability and applicability of ADAPT across a wide range of contexts.
The pruning process used by ADAPT results in heterogeneous context lengths, automatically determining the optimal prompt length at each layer, which is an improvement over manually designed prompts that tend to be homogeneous and less efficient.

**Weaknesses:**

ADAPT shows significant performance degradation in certain categories, such as the Pets class, where it fails to rank even in the top three. It is regrettable that the authors did not conduct further discussion and research on this issue.
The highly heterogeneous prompt lengths determined by the pruning mechanism could make the model harder to implement in practical scenarios where consistency and predictability are valuable, compared to using manually fixed homogeneous prompt lengths.
Although ADAPT optimizes both text and image branches independently, there is no explicit mechanism mentioned to ensure that the branches remain aligned in terms of context length adjustments. This could potentially lead to imbalances that affect the model's overall performance.

**Questions:**

Could the authors provide more details about the scoring function used to determine token importance during pruning? Were any alternative scoring mechanisms considered, and if so, why was the current approach chosen?
How does ADAPT ensure stability during the pruning process, especially given the highly heterogeneous prompt lengths across different layers? Are there any safeguards in place to avoid over-pruning, where the model could lose important contextual information?
The evaluation on 11 datasets showed varying degrees of performance, with some datasets exhibiting reduced accuracy compared to the baseline. Could the authors elaborate on the potential reasons behind these inconsistencies and suggest strategies that could mitigate these issues in future iterations of ADAPT?
Given the independence of the pruning processes for the text and image branches, is there any mechanism in place to maintain synchronization between the two branches during training? If not, could this lead to potential issues in multimodal understanding?

---

> ### Author Response · Authors · 2024-11-21
>
> The authors thank the reviewer for the feedback. Here is one-to-one response:
>
> - Weakness 1: Some datasets do not have the best performance
>
> We understand the reviewer’s concern and agree that this is an important issue to elaborate on. Despite the performance variances across datasets, our method achieves the best performance on 7 out of 11 datasets and ranks in the top 3 on 8 out of 11 of them. The second best method LAMM achieves the best performance on 0 out of 11 datasets and the top 3 performance on 7 out of 11 datasets. The third best method ProGrad achieves the best performance on 2 out of 11 datasets and the top 3 performance on 6 out of 11 datasets.
>
> We force the same $T_{target}$ to ensure a similar number of trainable parameters across datasets. When allowing different $T_{ target}$ for different datasets, we can have a significant performance gain on some datasets such as Food101 as shown in Table 1 of the updated manuscript. The details regarding Adapt (adaptive $T_{target}$) are reported in Appendix A.9.
>
> We also want to note that the “no-winner-takes-all” finding has been consistently observed in the study of continuous prompting methods for VLMs. When PLOT (ICLR 2023) [1] is published, there are two prompting method baselines. It achieves the best performance on 6 out of 11 datasets. When LAMM (AAAI 2024) [2] is published, there are two prompting method baselines. It achieves the best performance on 7 out of 11 datasets. Thus, none of the methods consistently rank within the top 3 on all the datasets
>
> On the Pets dataset, different methods have similar performance. Even the zeroshot CLIP can achieve reasonably good performance. We would like to draw attention to the fact that our method has pronouncedly better performance on challenging datasets such as DTD, EuroSAT and Aircraft where the zeroshot CLIP has poor performance.
>
> - Weakness 2: Heterogeneous prompt lengths could make the model harder to implement in practical scenarios
>
> Our method uses $T_{target}$ as a hyperparameter to automatically determine the context lengths at different depths by pruning (please refer to Algorithm 1 line 13). The manually designed deep prompting method has a hyperparameter of prompting depth. Hence, different depths can have different context lengths (when the depth is equal to or smaller than d, the context length is t per layer; when the depth is larger than d, the context length is 0 per layer). Our method intentionally introduces heterogeneous context lengths and finds that it achieves better performance than manually designed homogeneous context lengths.
>
> When we report the performance, it is averaged over 3 runs. We included the standard deviation in Appendix A.6.
>
> Finally, to fully address the reviewer’s comment, could the reviewer elaborate on the difficulty in practical scenarios in terms of consistency and predictability?
>
> - Weakness 3: There is no explicit mechanism to ensure the two branches are aligned
>
> This is a great suggestion for an ablation study. In Adapt, we intentionally enable the different total context lengths in the text and image branches. Therefore, the setting of the same total context length in text and image branches is included in our search space of the current setting. Following the reviewer’s suggestion, we conducted experiments that ensured the same context lengths in the image and text branches. The modification we made was that instead of combining scores in text and image branches, we ranked scores in the text branch and scores in the image branch. In each pruning step, we pruned r_p tokens in the text branch and r_p tokens in the image branch. We noticed a pronounced performance degradation by applying constraints on the total context length of two branches to be the same (Adapt Constraint). The results comparison is:
>
> | Method | Caltech101 | DTD | EuroSAT | Aircraft | Food101 | Flowers | Pets | Cars | Sun | UCF | ImageNet | Average |
> | --- | --- | --- | --- | --- | --- | --- | --- | --- | --- | --- | --- | --- |
> | Adapt w/   constraint | 95.43 | 69.17 | 81.50 | 49.13 | 83.30  | 98.03 | 87.47 | 84.13 | 73.80 | 83.37| 69.93 | 79.57 |
> | Adapt w/o constraint | 95.63 | 72.03 | 92.53 | 50.93 | 83.47 | 97.97 | 91.07 | 86.17 | 73.67 | 84.40  | 70.83 | 81.70 |
>
> The experiment indicates that two branches do not need to be aligned. Less constraint is beneficial to fully exploit the power of prompt tuning. Please refer to Appendix A.11 for more details.
>
> - Question 1: more details on the scoring function
>
> In our initial submission, we tested Snip, Gradient norm and l2-norm as scoring functions (shown in Eq. 6 and the ablation study “Score computation” in Sec. 4.3). We empirically found that Snip works better. Snip considers gradients and magnitudes of model parameters. Gradient norm only considers gradients. l2-norm only considers magnitudes of model parameters.

---

> > ### Author Response · Authors · 2024-11-21
> >
> > - Question 2: How to ensure stability? Are there safeguards?
> >
> > Even using fixed $T_{target}$, our method can surpass the baseline methods. Further, we can use the validation accuracy to determine the optimal $T_{target}$ for each dataset. The performance of using adaptive $T_{target}$ is reported in Appendix A.9.
> >
> > When we applied pruning, only inserted soft prompts were pruned while the original embeddings were not pruned. Taking an extreme case as an example, if all soft prompts at a certain depth are pruned, that indicates the prompt depth decreases by one.
> >
> > - Question 3: Potential reasons for the inconsistency of the performance
> >
> > We noticed that there is no method that performs uniformly well across all datasets. To be fair to the baselines, we used the same hyperparameters including $T_{target}$ for all datasets. However, as shown in Table 2, when there is a relatively large distribution shift (e.g. EuroSAT dataset contains satellite images), a larger $T_{target}$ can lead to a better performance. When the distribution shift is small (e.g. Caltech101 dataset contains generic images), a smaller $T_{target}$ leads to better performance. Hence, one potential reason causing the different ranks on different datasets is that different datasets, depending on how different they are from the pre-trained dataset, might require different prompt complexity. We also included an adaptive selection method of Adapt in  Appendix A.9 to select the best $T_{target}$ for each dataset. The results show that the average accuracy can be improved compared to using Adpat with a universal and fixed $T_{target}$. To make a fair comparison, we use the $T_{target}$ to apply the constraint on the prompt complexity.
> >
> > Rank results; We use a fixed $T_{target}$, not an optimal hyperparameter. The reason we use fixed hyperparameters is to make a fair comparison.
> >
> > - Question 4: synchronization between two branches and potential issue in multimodal understanding
> >
> > We would like to point out that we are not pruning the original tokens but only the additional inserted context tokens. Thus, the original tokens are not pruned. The inserted tokens are trained such that the overall loss of the model on the given task is reduced. Thus, even when we prune the number of additional tokens, the loss of the model is still reduced indicating that its performance is not hampered but only improved. We would also like to point out that some works (e.g. CoCoOp [5] and ProGrad [6]) only insert prompts to text branches, which can boost the performance of the pre-trained model.
> >
> > [1] Chen, Guangyi, et al. "Plot: Prompt learning with optimal transport for vision-language models."International Conference on Learning Representations. 2023.
> >
> > [2] Gao, Jingsheng, et al. "LAMM: Label Alignment for Multi-Modal Prompt Learning." Proceedings of the AAAI Conference on Artificial Intelligence. Vol. 38. No. 3. 2024.
> >
> > [3] Khattak, Muhammad Uzair, et al. "Maple: Multi-modal prompt learning." Proceedings of the IEEE/CVF Conference on Computer Vision and Pattern Recognition. 2023.
> >
> > [4] Zang, Yuhang, et al. "Unified vision and language prompt learning." arXiv preprint arXiv:2210.07225 (2022).
> >
> > [5] Zhou, Kaiyang, et al. "Conditional prompt learning for vision-language models." Proceedings of the IEEE/CVF conference on computer vision and pattern recognition. 2022.
> >
> > [6] Zhu, Beier, et al. "Prompt-aligned gradient for prompt tuning." Proceedings of the IEEE/CVF International Conference on Computer Vision. 2023.

---

> > ### Author Response · Authors · 2024-11-27
> >
> > Dear reviewer,
> >
> > We appreciate the time and efforts the reviewer dedicated to evaluating our work. As the deadline for the author-reviewer discussion is closing soon but we still have not received any feedback, we kindly ask if the reviewer could review our responses and let us know if any additional clarifications or modifications are required from our side.
> >
> > Authors

---

> > > ### Author Response · Authors · 2024-12-01
> > >
> > > We thank the reviewer for the time and effort spent in evaluating our work, As the author-reviewer discussion phase is ending soon, we have not had a chance to engage with the reviewer.  We would like to kindly ask if our response solves all of the reviewer’s concerns.

---

> ### Comment · Area_Chair_yWcZ · 2024-11-25
>
> Dear Reviewer Tr3E,
>
> Could you kindly review the rebuttal thoroughly and let us know whether the authors have adequately addressed the issues raised or if you have any further questions.
>
> Best,
>
> AC of Submission8374

---

### Meta-Review · Area_Chair_yWcZ · 2024-12-20

**Metareview:**

(a) The paper proposes a deep continuous prompting method called Adapt to address the limitations of conventional deep prompt tuning. Adapt encourages heterogeneous context lengths by automatically determining and pruning context tokens based on saliency scores.

(b) Strengths: The paper is well-written and presents a clear explanation of the proposed method. Extensive experiments on 11 downstream datasets demonstrate the effectiveness of Adapt. The idea of adding masks to prompts at different depths is an interesting and innovative approach.

(c) Weaknesses: The paper lacks clarity on how to set the number of pruning steps. The excessive use of mathematical symbols, particularly in Algorithm 1, hinders readability and should be improved. The introduction section is too brief and could benefit from further elaboration and paragraph breaks. The paper does not sufficiently discuss its relationship with dynamic neural networks, the convergence of model training, or the impact of setting T_target as a fixed value. Additionally, the analysis of learned binary masks and a more comprehensive comparison with other training settings and methods, such as UPT, is needed.

(d) The most important reasons for reject are: This work fails to compare/discuss several important baselines (e.g., PromptSRC[1]) in this field and the AC finds that there some incorrect performance shown in the tables/figures. First, this paper seems to deliberately ignore some strong sota baseline, e.g., PromptSRC[1]. The performance of PromptSRC in few-shot setting with 16-shot is 82.87, whilst the performance of the method in this work is 82.67. We suggest that the authors at least mention these baselines, rather than completely ignoring them. If the performance does not outperform, could the authors' method be applied on top of these baselines? Second, the AC discovers that there are inconsistencies between the performance reported in this work (MaPLe performance in Figure 7) and the published one [1] (MaPLe performance in Table 13). For the few-shot performance of MaPLe from 1-shot to 16-shot, this work reports around 65.5, 70, 73, 75.5, 78 from the Figure 7; whilst [1] reports 69.27,72.58, 75.37, 78.89, 81.79. The settings (few-shot 1, 2, 4, 8, 16) and methods (MaPLe) are identical, so the AC is unclear about the reason for such a large gap.

[1] Self-regulating Prompts: Foundational Model Adaptation without Forgetting, ICCV 23 (Google Citation 100+).

**Additional Comments On Reviewer Discussion:**

(a) Reviewer Tr3E points out that ADAPT shows significant performance degradation in certain categories, such as the Pets class, and lacks further discussion on this issue. The highly heterogeneous prompt lengths resulting from the pruning mechanism could reduce the model's practicality due to the need for consistency and predictability. Additionally, the lack of an explicit alignment mechanism between the text and image branches may lead to imbalances, potentially affecting the model's overall performance. The reviewer fails to respond during the discussion phase.

(b) Reviewer qkws highlights that while adding learnable masks to prompts is an interesting idea, it is similar to existing methods and lacks a discussion on the differences with related work. The paper does not explore the effect of larger T_target values on performance, and the upper bound of the proposed method remains unclear. Additionally, the reviewer suggests conducting ablations on prompt depth and context length and providing results for various few-shot training settings to better demonstrate the method's effectiveness. The reviewer actively participates in the discussion phase and raises the score from 3 to 5, stating there is still room for improvement (e.g., the method effectiveness in 1/2/4/8 shot settings, and more related works in 'Sparse Training' section).

(c) Reviewer bqgw points out that the convergence of model training is overly reliant on a fixed T_target, which may not be suitable for all datasets, and could lead to issues with mask sparsity in the image encoder. The paper lacks a detailed analysis of the learned binary masks, particularly regarding discrepancies in context token requirements across layers. Additionally, the reviewer suggests including results for various few-shot training settings and comparing ADAPT with the UPT method for a more comprehensive evaluation. The rebuttal addresses most of the concerns. The reviewer finds that automating the selection of T_target would be a valuable improvement thus maintains the initial score of 5.

(d) Reviewer Xqnt suggests that the paper lacks clarity in setting the number of prune steps (rp) and in the excessive use of mathematical symbols, particularly in Algorithm 1, which hampers readability. The introduction should be expanded into more paragraphs, and the authors should discuss and cite related work on dynamic neural networks. Additionally, the reviewer recommends improving the overall presentation and labeling the proposed method as "Adapt (Ours)" in Figure 1. The questions are well addressed and the reviewer raises the score to 6.

---

### Decision · Program_Chairs · 2025-01-22

Reject